

# Neoproterozoic and post-Caledonian exhumation and shallow faulting in NW Finnmark from K/Ar dating and p/T analysis of fault-rocks

5    Jean-Baptiste P. Koehl[1,2], Steffen G. Bergh[1,2], Klaus Wemmer[3]

[1]Department of Geosciences, University of Tromsø, N-9037 Tromsø, Norway.

[2]Research Center for Arctic Petroleum Exploration (ARCEx), University of Tromsø, N-9037 Tromsø, Norway.

[3]Geoscience Centre, Georg-August-University Göttingen, Goldschmidtstraβe 3, 37077 Göttingen, Germany.

*Correspondence to*: Jean-Baptiste Koehl (jean-baptiste.koehl@uit.no)

**Abstract**. Well-preserved fault gouge along brittle faults in Paleoproterozoic, volcano-sedimentary rocks of the Raipas Group exposed in the Alta-Kvænangen tectonic window in northern Norway yielded latest Mesoproterozoic (ca. 1050 ± 15 Ma) to mid Neoproterozoic (ca. 825-810 ± 18 Ma) K/Ar ages. Pressure-temperature estimates from microtextural and mineralogy analyses of fault-rocks indicate that brittle faulting may have initiated at depth of 5-10 km during the opening of the Asgard Sea in the latest Mesoproterozoic-early Neoproterozoic (ca. 1050-945 Ma), and continued with a phase of shallow faulting during to the opening of the Iapetus Ocean-Ægir Sea and the initial breakup of Rodinia in the mid Neoproterozoic (ca. 825-810 Ma). The predominance and preservation of synkinematic smectite and subsidiary illite in cohesive and non-cohesive fault-rocks indicate that Paleoproterozoic basement rocks of the Alta-Kvænangen tectonic window remained at shallow crustal levels (< 3.5 km) and were not reactivated since mid Neoproterozoic times. Slow exhumation rate estimates for the early-mid Neoproterozoic (ca 10-75 m per Ma) suggest a period of tectonic quiescence between the opening of the Asgard Sea and the breakup of Rodinia. In the Paleozoic, basement rocks in NW Finnmark were overthrusted by Caledonian nappes along low-angle thrust detachments during the closing of the Iapetus Ocean-Ægir Sea. K/Ar dating of non-cohesive fault-rocks and microtexture-mineralogy of cohesive fault-rock truncating Caledonian nappe units show that brittle (reverse) faulting potentially initiated along low-angle Caledonian thrusts during the latest stages of the Caledonian Orogeny in the Silurian (ca. 425 Ma) and was accompanied by epidote/chlorite-rich, stilpnomelane-bearing cataclasite (type 1) indicative of a faulting depth of 10-16 km. Caledonian thrusts were inverted (e.g. Talvik fault) and





later truncated by high-angle normal faults (e.g. Langfjord-Vargsund fault) during subsequent, late Paleozoic, collapse-related widespread extension in the Late Devonian-early Carboniferous (ca. 375-325 Ma). This faulting period was accompanied by quartz- (type 2), calcite- (type 3) and laumontite-rich cataclasites (type 4), which crosscutting relationships indicate a progressive
exhumation of Caledonian rocks to zeolite facies conditions (i.e. depth of 2-8 km). An ultimate period of minor faulting occurred in the late Carboniferous-mid Permian (315-265 Ma) and exhumed Caledonian rocks to shallow depth 1-3.5 km. Alternatively, late Carboniferous (?) - early/mid Permian K/Ar ages may reflect late Paleozoic weathering of the margin. Exhumation rates estimates indicate rapid Silurian-early Carboniferous exhumation and slow exhumation in the
late Carboniferous-mid Permian, supporting decreasing faulting activity from the mid-Carboniferous. NW Finnmark remained tectonically quiet in the Mesozoic-Cenozoic.

## 1. Introduction

Onshore and nearshore areas of Finnmark and shallow parts of the Barents shelf, such as the Finnmark Platform, are underlain by Archean-Paleoproterozoic basement rocks exposed onshore in coastal ridges (Zwaan, 1995; Bergh et al., 2010) and tectonic windows (Reitan, 1963; Roberts, 1973; Zwaan & Gautier, 1980; Gautier et al., 1987; Bergh & Torske, 1988; Jensen, 1996) of the overlying Caledonian nappe stack (Roberts, 1973; Corfu et al., 2014). These basement rocks
are part of the Fennoscandian Shield onto which Caledonian nappes were overthrusted in the Silurian (Townsend, 1987; Corfu et al., 2014; Figure 1). Post-Caledonian extension started in the Devonian (Roberts et al., 2011; Davids et al., 2013; Koehl et al., 2018) with reactivation of Proterozoic and Caledonian ductile fabrics and continued through the mid Permian and, later on, through the Mesozoic and early Cenozoic when multiple brittle faults onshore and nearshore and
major offshore rift basins formed prior to the opening of the NE Atlantic Ocean (Breivik et al., 1995; Gudlaugsson et al., 1998; Bergh et al., 2007; Faleide et al., 2008; Indrevær et al., 2013; Koehl et al., 2018).

Critical to the understanding of post-Caledonian extension and brittle fault evolution is the nature and timing of faulting. We have dated multiple brittle faults using K-Ar method of non-
cohesive fault rocks (Lyons & Snellenburg, 1971) along several major brittle faults in NW Finnmark, along fault segments of the Trollfjorden-Komagelva Fault Zone (TKFZ; Siedlecka &



Siedlecki, 1967; Siedlecki, 1980; Herrevold et al., 2009; Figure 1) and Langfjorden-Vargsundet fault (LVF; Zwaan & Roberts, 1978; Lippard & Roberts, 1987; Figure 1). The TKFZ, which crops out in northern (Koehl et al. submitted) and eastern Finnmark (Siedlecki, 1980), represents a major

Neoproterozoic fault zone that was active through various episodes of (Timanian and Caledonian) transpression and subsequent extension (Herrevold et al., 2009), whereas the LVF corresponds to a large, zigzag-shaped, NE-SW trending, margin-parallel fault complex which age is yet uncertain (Zwaan & Roberts, 1978; Roberts & Lippard, 2005; Koehl et al., submitted). This fault, however, extends onto the Finnmark Platform east where it bounds a triangular-shaped, half-graben basin of

presumed Carboniferous age (Figure 1; Koehl et al., 2018).

The main goal of this work is to constrain the timing of brittle fault initiation (Proterozoic and/or post-Caledonian), and to discuss the reactivation and exhumation history of faults in basement rocks and Caledonian units of NW Finnmark,. We focus on margin-parallel brittle faults in basement rocks of the Alta-Kvænangen tectonic window in the Altafjorden area, for comparison

with exposed fault segments of the Neoproterozoic, margin-oblique TKFZ (Figure 1). We also mapped and analyzed post-Caledonian fault segments and splays of the margin-parallel LVF to compare with the age of similar, offshore, basin-bounding faults (e.g. Troms-Finnmark and Måsøy fault complexes; Figure 1) and associated syn-tectonic sedimentary rocks on the Finnmark Platform and in offshore basins, e.g. the Hammerfest and Nordkapp basins (Gabrielsen et al. 1990;

Indrevær et al., 2013).

A second goal is to evaluate the amount of exhumation the margin underwent from the Neoproterozoic to the Caledonian Orogeny and from the collapse of the Caledonides to present times. Thus, we sampled cataclastic fault-rocks along multiple brittle faults including brittle faults that crosscut Archean-Paleoproterozoic rocks in fresh road cuts in Altafjorden, and faults like the

LVF and TKFZ in Caledonian thrust nappe rocks (Figure 1). Then, we analyzed characteristic mineral assemblages for each cataclastic fault-rock, i.e. each faulting event recorded, which we used in conjunction with crosscutting relationships between fault-rocks to reconstruct the evolution of p/T conditions (i.e. depth) during faulting and, thus, resolve the exhumation history of the margin. We compare our results with those from analog studies in Western Troms (Davids et al.,

2013; Indrevær et al., 2014; Davids et al., submitted) and Finnmark (Torgersen et al., 2014), and discuss regional implications for the tectonic evolution of the Troms-Finnmark margin during post-Caledonian extension.



## 2. Geological setting


### 2.1. Precambrian basement rocks and Caledonian nappes

The bedrock geology of NW Finnmark consists of Archean-Paleoproterozoic metavolcanic and metasedimentary rocks that occur in tectonic windows, e.g. the Alta-Kvænangen (Bøe & Gautier, 1978; Zwaan & Gautier, 1980; Gautier et al., 1987; Bergh & Torske, 1988), Altenes (Jensen, 1996) and Repparfjord-Komagfjord tectonic windows (Reitan, 1963; Pharaoh et al., 1982, 1983), overlying Caledonian nappes, e.g. Kalak Nappe Complex (Ramsey et al., 1979, 1985; Kirkland et al., 2005) and Magerøy Nappe (Andersen, 1981, 1984), and intruded by igneous rocks of the Seiland Igneous Province (Elvevold et al., 1994; Pastore et al., 2016) and Honningsvåg Igneous Complex (Corfu et al., 2006). Precambrian basement rocks are variably metamorphosed but generally show greenschist facies mineral assemblages in the study area (Bøe & Gautier, 1978; Zwaan & Gautier, 1980; Bergh & Torske, 1988).

The Caledonian Kalak Nappe Complex is thought to represent a Laurentia-derived unit that was thrusted over Precambrian basement rocks of Baltica during the Caledonian Orogeny (Kirkland et al., 2008). Metasedimentary rocks of the Kalak Nappe Complex were metamorphosed to amphibolitic facies conditions and are composed of metapsammites, schists and paragneisses crosscut by low-angle thrusts and shear zones (Ramsey et al., 1979; 1985). The Kalak Nappe Complex is intruded by mafic and ultramafic rocks of the Seiland Igneous Province (Elvevold et al., 1994) that form two deep roots below the island of Seiland and Sørøya (Pastore et al., 2016). The structurally overlying Magerøy Nappe is made of tightly folded, greenschist facies, metasedimentary rocks that are separated from the Kalak Nappe Complex by a low-angle thrust onshore Magerøya (Andersen, 1981, 1984). This nappe unit was intruded by mafic rocks of the Honningsvåg Igneous Complex during the Caledonian Orogeny (Corfu et al., 2006).

### 2.2. Brittle faulting and previous age dating in North Norway

#### 2.2.1. Brittle faults trends



Precambrian basement rocks and Caledonian nappes in coastal areas of northern Norway are truncated by several major brittle faults and fracture sets, striking NNE-SSW, ENE-WSW and

WNW-ESE (Bergh et al. 2007; Eig & Bergh 2011; Indrevær et al. 2013; Koehl et al. submitted) and of presumed post-Caledonian age. The timing of formation of these faults is uncertain and yet unresolved, but most are thought to be post-Caledonian (Davids et al., 2013, submitted), although Precambrian ages cannot be excluded (Torgersen et al., 2014; Koehl et al., submitted). The three fault sets commonly form two major fault systems. On the one hand, NNE-SSW and ENE-WSW

faults often interact to produce zigzag-shaped fault complexes. An example in NW Finnmark is the LVF, which is made up of alternating ENE-WSW and NNE-SSW trending fault segments (Zwaan & Roberts, 1978; Lippard & Roberts, 1987; Koehl et al. submitted) and resembles the offshore, basin-bounding, zigzag-shaped Troms-Finnmark Fault Complex in map-view (Figure 1; Indrevær et al. 2013). This resemblance is verified by offshore prolongation of the LVF into shallow

Carboniferous half-graben on the Finnmark Platform east (Koehl et al., 2018). On the other hand, WNW-ESE striking faults are usually observed as swarms of high-frequency, (sub-) parallel fractures. A key example is the TKFZ, a major Neoproterozoic fault zone that extends from the Varanger Peninsula in the east and crops out onshore the island of Magerøya in the west (Figure 1). This fault complex is made up with multiple segments of sub-parallel, WNW-ESE trending

brittle faults that die out just west of Magerøya (Koehl et al. submitted). Several of these fault segments were intruded by highly magnetic dolerite dykes during an early Carboniferous extensional event (Roberts et al. 1991; Lippard & Prestvik 1997; Nasuti et al. 2015), when the TKFZ acted as a strike-slip transfer fault and segmented onshore-nearshore areas of NW Finnmark from the offshore Finnmark Platform east during late/post-Caledonian extension (Koehl et al.,

2018, submitted).

### 2.2.2. *Dating of brittle faults in North Norway*

Previous K-Ar dating of synkinematic illite/muscovite in fault gouge in NW Finnmark show that brittle faults in the Repparfjord-Komagfjord tectonic window, like the Kvenklubben

fault, formed in Precambrian times and were reactivated as Caledonian thrust and, later on, as normal faults during late/post-orogenic extension and subsequent rifting (Torgersen et al. 2014). Non-cohesive fault-rocks sampled by Torgersen et al. (2014) showed enrichment in authigenic smectite and chlorite clay minerals and a rather low content of illite/muscovite.



Farther southwest along the margin, in Western Troms, similar coastal brittle faults display
cohesive cataclastic fault-rocks with epidote, chlorite and pumpellyite (Indrevær et al. 2014), and
these faults are juxtaposed with amphibolite facies Precambrian rocks of the West Troms Basement
Complex (Zwaan 1995; Bergh et al. 2010). These faults are interpreted to have formed during late
Paleozoic extension at depth > 10 km and to have been exhumed to shallower crustal level < 8.5
km, as shown by the widespread occurrence of pumpellyite mineral in fault-rocks. These faults
yielded late Paleozoic K/Ar ages and are possibly associated with post-Caledonian extension in the
Devonian-Carboniferous (Davids et al. 2013).

## 3.  Methods

### 3.1. Structural field data

In summer 2015, we acquired extensive structural field data along brittle faults in NW
Finnmark, which we compiled, interpreted and discussed in earlier contributions (Bergø, 2016;
Lea, 2016; Koehl et al., submitted). Among the numerous brittle faults cropping out in NW
Finnmark, we selected ten based on their proximity to major faults (e.g. LVF, TKFZ), location
relative to major faults (footwall/hanging-wall) and according to their strike (parallel to dominant
fault trends in Finnmark), which we briefly describe from a structural perspective at outcrop scale.
Fault geometries and kinematic indicators will be used in conjunction with cohesive and non-
cohesive fault-rock composition and with the result of K/Ar dating of fault gouge to propose an
evolutionary model for the tectonic evolution and exhumation history of the SW Barents Sea
margin and NW Finnmark.

### 3.2. Microscopic analysis of onshore cohesive fault-rock

We collected cohesive fault-rock samples along numerous brittle faults we encountered in
NW Finnmark (including the ten dated faults), and we used them to investigate kinematic indicators
along the selected brittle faults at microscale. We also studied mineral assemblages included in
brittle fault-rocks in order to constrain metamorphic facies (p/T) conditions during faulting,
therefore adding to the understanding of the exhumation and uplift history of the SW Barents Sea



margin. When needed, thin sections were analyzed through an optical microscope and a Scanning
Electron Microscope (SEM) at the University of Tromsø to obtain more detailed information about
mineral composition.

### 3.3. K/Ar dating and mineralogical analysis of fault gouge

We sampled non-cohesive fault-rock along brittle faults in NW Finnmark and attempted to
date authigenic (i.e. synkinematic) illite clay mineral formed during faulting events (e.g. Vrolijk &
van der Pluijm, 1999; Davids et al., 2013; Torgersen et al., 2014; Ksienzyk et al., 2016), which
platy crystal shape differs from their irregular detrital counter-part (e.g. Davids et al., 2013;
Torgersen et al., 2014). The ten dated samples of non-cohesive fault-rock are referred to as sample
1-10 in the text and figures. K/Ar dating of fault gouge was carried out in the K/Ar laboratory
facility at the University of Göttingen, Germany. Three grainsize fractions were analyzed for each
sample: "2-6 µm", "< 2 µm" and "< 0.2 µm". Clay-rich fault gouge samples were resolved in water
and wet-sieved using a 63 µm sieve. The fraction < 63µm was used to extract the clay fractions <
2µm by settling in Atterberg cylinders. The fractions < 0.2µm have been separated using an ultra-
centrifuge. All these fine fractions were examined (XRD) for mineralogical composition and
determination of the illite crystallinity using a PHILIPS PW 1800 diffractometer

Illite crystallinity, the peak width at half height of the 10-Å peak, was determined using a
computer program developed at the University of Göttingen. Digital measurement of illite
crystallinity was carried out by step scan (301 points, 7-10° 2Θ, scan step 0.010° 2Θ, integration
time 4 s, receiving slit 0.1mm, automatic divergence slit). Illite crystallinity determinations have
shown to be a sensitive indicator for the degree of very low grade metamorphism in clastic
sediments. Reviews of the preparation techniques and the interpretation have been given for
example by Kisch (1991) and Krumm (1992). All samples have been investigated in duplicates (A
and B). The measurements were carried out in the „air dry" and the „ethylene glycol saturated"
status in order to detect expandable layers of smectite type minerals. Smectite classification
(Reichweite) has been determined following Moore & Reynolds (1997). Illite crystallinity is
expressed as Kübler Index in (KI, Δ°2Θ), the limits for diagenesis/anchizone (ca. 200°C) and
anchizone/epizone (300°C) are 0.420° and 0.250° Δ°2Θ (Kübler 1967, 1968, 1984), respectively.



The argon isotopic composition was measured in a pyrex glass extraction and purification line coupled to a Thermo Scientific ARGUS VI ™ noble gas mass spectrometer operating in static mode. The amount of radiogenic $^{40}$Ar was determined by isotope dilution method using a highly enriched $^{38}$Ar spike from Schumacher, Bern (Schumacher, 1975). The spike is calibrated against the biotite standard HD-B1 (Fuhrmann et al., 1987). The age calculations are based on the constants

recommended by the IUGS quoted in Steiger & Jäger (1977). Potassium was determined in duplicate by flame photometry using a BWB-XP flame photometer ™. The samples were dissolved in a mixture of HF and $HNO_3$ according to the technique of Heinrichs & Herrmann (1990). The analytical error for the K/Ar age calculations is given on a 95% confidence level (2σ). Details of argon and potassium analyses for the laboratory in Göttingen are given in Wemmer (1991).


*Temperature constraints from illite-smectite clay minerals*

Since the dominant synkinematic clay mineral in the analyzed fault-rocks are smectite and subsidiary interlayered illite-smectite clay, we use the smectite-illite clay mineral reaction to infer maximum/minimum temperature estimates for faulting events in NW Finnmark (Eberl et al. 1993;

Huang et al. 1993; Morley et al. 2018). Synkinematic illite often grows due to illitisation of smectite and, alternatively, due to dissolution-precipitation of existing clay minerals of the bedrock (Vrolijk & van der Pluijm, 1999). Illitisation along fault surfaces is enhanced by temperature increase, e.g. related to frictional heating, hydrothermal processes or burial, grain comminution, strain, changes in fluid composition and fluid/rock ration (Vrolijk & van der Pluijm, 1999). Illitisation of smectite

is commonly thought to begin at a temperature range of 40-70°C (Jennings & Thompson, 1986; Harvey & Browne, 2000; Ksienzyk et al., 2016).

*Interpretation of inclined age spectra*

K/Ar dating requires targeted minerals to behave as "closed systems" with no loss of argon

or potassium (Lyons & Snellenburg, 1971). Mineral closure temperature varies with grainsize and is lower for finer grains. For example, aggregates of fine grains may accidentally be incorporated and dated as part of coarser fractions of fault gouge, thus leading coarse fractions to yield younger ages than finer fractions (Hamilton et al., 1989; Heizler & Harrison, 1991). More specifically along shallow faults, illite grains < 2 µm crystallize below the closure temperature of the K-Ar system (>

250°C; Velde, 1965) and, thus, provide with robust, synkinematic crystallization ages rather than

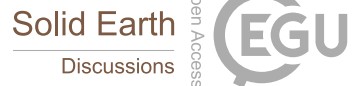


less accurate, generally younger cooling ages obtained along deeper faults (Hunziker et al., 1986; Ksienzyk et al., 2016). Further, contrarily to metamorphism-related heating, the short heating time associated with hydrothermal events or frictional heating along brittle faults are unlikely to reset illite ages, which would require longer exposure to temperature > 250°C (Torgersen et al., 2014),

thus suggesting that K/Ar ages on illite along shallow faults provide reasonable estimates of the age of faulting.

Mixing of host-rock inherited minerals, e.g. detrital illite-muscovite, with authigenic illite may influence K/Ar ages and cause age dispersion in faults, notably in the coarser fractions dated (Hower et al., 1963; Vrolijk & van der Pluijm, 1999). However, inherited illite-muscovite may be

distinguished from authigenic clay minerals as they display more irregular shapes than their generally platy authigenic counter-parts (e.g. Torgersen et al., 2014). In addition, faulting may even isotopically reset fine-grained, hostrock illite-muscovite, thus yielding ages bearing no influence of inherited, older minerals (Vrolijk & van der Pluijm, 1999). In addition, fault-inherited illite may also affect K/Ar ages, which is especially verified along repeatedly active, progressively exhumed

faults because high-temperature illite may survive low-temperature reactivation of the faults (Davids et al., 2013; Viola et al., 2013). Another mineral that may have a significant impact on K/Ar ages is host rock-inherited K-feldspar. Most importantly, K-feldspar has a significantly lower closure temperature (350-150°C) than illite clay mineral (> 250°C), hence yielding younger ages than the actual age of faulting, particularly for finer fractions in fault gouges (Lovera et al., 1989).

Hornblende may also affect K/Ar dating of illite (Torgersen et al., 2015) but was not encountered in any fault-rock samples in NW Finnmark and its effect are therefore not considered here.

## 4. Results

We sampled brittle fault rocks for K-Ar dating and microtextural analysis from several dominant fault systems and fault trends in NW Finnmark (Figure 1; Koehl et al., submitted). The sampling sites include (i) faults in Paleoproterozoic rocks of the Raipas Group (Zwaan & Gautier, 1980) in Altafjorden (samples 3 & 4), (ii) faults in the Caledonian Kalak Nappe Complex along segments and splay-faults of the LVF (samples 1, 2, 5, 6 & 7), and (iii) faults in rocks of the Kalak

Nappe Complex and Magerøy Nappe (sample 8, 9 & 10) adjacent to segments of the TKFZ (Figure 1). For the sampled faults, we first describe trends, field relations and kinematics of the faults.



Second, we describe the mineral assemblages and microstructures of sampled cohesive fault-rocks in order to infer deformation mechanisms and estimate the p-T conditions during faulting and exhumation. Third, we present the K-Ar data and mineralogical results obtained on fault gouge.


## 4.1. Field relations of sampled brittle faults in NW Finnmark

### 4.1.1. *Brittle faults in Paleoproterozoic basement rocks (samples 3 & 4)*

Samples 3 and 4 (Figure 2a & b) are from the Altafjorden fault 1 and 2, NW-dipping brittle faults in new, fresh road-cuts along the western shore of Altafjorden, truncating meta-arkoses of the Paleoproterozoic Raipas Group (Skoadduvarri Sandstone) of the Alta-Kvænangen tectonic window (Zwaan & Gautier, 1980; Bergh & Torske, 1986, 1988). The two sampled faults are located a few tens of meters away from each other, display m-thick fault-cores mostly composed of grey-colored clay particles in non-cohesive fault gouge and partly cohesive cataclasites (Figure 2a & b). The Altafjorden fault core also show multiple slip surfaces cemented by cm-thick quartz grains (Figure 2a). Slickenside lineations on fault surfaces indicate normal dip-slip movement. A mafic band is bent into (drag folded) the fault core facilitating normal down-to-the NW sense of shear (Figure 2a). The absence of this mafic bed in the footwall of the Altafjorden fault suggests that the fault accommodated vertical displacement > 5-6 m.


### 4.1.2. *Brittle faults along the LVF (samples 1, 2, 5, 6 & 7)*

We sampled fault-rocks along several segments/splay-faults of the LVF (Torgersen et al., 2014; Koehl et al., submitted). This regional fault complex can be traced from Sørkjosen in the south (cf. location of samples 1 & 2 in Figure 1) to the Porsanger Peninsula in the north (Figure 1) and defines a zigzag-shaped pattern of alternating, NNE-SSW to ENE-WSW trending fault segments that dominantly dip WNW and NNW, respectively (Koehl et al., submitted). Samples 1 and 2 are taken from two minor NW-dipping fault splays (Figure 2c & d) in the footwall of the Sørkjosen fault, a major fault-segment of the LVF (Figure 1; Koehl et al., submitted). The faults crosscut granodioritic gneisses of the Kalak Nappe Complex and display thin, 10-40 cm-thick fault-cores with lenses of dark clayish gouge material (Figure 2c & d). The northern of these faults accommodates ca. 5-8 m top-to-the-NW, normal displacement of a 20-30 cm-thick layer of mafic amphibolite (Figure 2c), while the southern fault offsets the same mafic layer by ca. 2 m top-to-



the-NW (Figure 2d). Slickenside lineations along these two faults support normal dip-slip sense of shear (Figure 2c & d).

Along strike northeastward, we sampled another subsidiary fault linked to the LVF along the western shore of Altafjorden, the Talvik fault (sample 5; Figure 1), a low-angle, N-dipping fault that crosscuts arkosic meta-psammites of the Kalak Nappe Complex (Figure 2e). The Talvik fault shows evidence of both brittle and ductile faulting in a ca. one meter-thick fault-core made of semi-ductile, mylonitic fault-rock, overprinted by calcite- and quartz-bearing cataclasite, as well as thin

layers of non-cohesive fault gouge along distinct fault surfaces (Figure 2e). Field observations of quartz-sigma clasts and S-C fabrics in the mylonites reveal top-to-the-south thrusting, whereas slickengrooves and asperities are present along distinct brittle fault surfaces, implying top-to-the-N motion along the Talvik fault (Figure 2e). Brittle offset of ductile quartz sigma-clasts confirms that brittle fabrics are younger than ductile fabrics.

Sample 6 was taken along a high-angle, NNE-SSW trending and WNW-dipping brittle fault that crops out along the eastern shore of Altafjorden (cf. Figure 1 and Figure 2f). This fault defines the southeastern boundary of a graben structure in garnet-rich meta-psammite of the Kalak Nappe Complex and is characterized by a ca. one meter-thick fault-core made of non-cohesive clayish fault gouge and adjacent epidote-rich cataclasite (Figure 2f). Slickenside lineations indicate down-

to-the-WNW, dip-slip normal movement, which is consistent with normal dip-slip offsets of boudinaged mafic dykes across nearby NNE-SSW trending faults (Figure 2f).

The final fault-rock sample along LVF segments and splays (sample 7; Figure 1) is from the steep, NNE-SSW trending, SE-dipping Snøfjorden-Slatten fault on the Porsanger Peninsula (cf. Figure 1 and Figure 2g; Passe, 1978; Townsend, 1987b), which represents a major, antithetic fault

segment/splay of the LVF (Koehl et al., submitted). This fault crosscuts felsic metasedimentary rocks and micaschists of the Kalak Nappe Complex and displays a several meter-wide fault-core that is made of non-cohesive iron- and quartz-rich fault gouge, including a few lenses of cohesive fault-rock (Figure 2g). Slickensided fault surfaces reveal down-to-the-SE, normal dip-slip movement, probably a few meters to a few tens of meters due to the presence of the same host rock

on both sides of the fault (Figure 2g).

### 4.1.3.   Brittle faults adjacent to the TKFZ (samples 8, 9 & 10)



Subvertical, WNW-ESE trending brittle faults and fracture systems are widespread near
and within Magerøy Nappe units on the Porsanger Peninsula and the island of Magerøya (Figure
1; cf. Koehl et al. submitted). Sample 8 is from fault gouge found along an anomalously low-angle,
WNW-ESE striking, NNE-dipping fault on the Porsanger Peninsula (Figure 1 and Figure 2h). The
fault crosscuts garnet-mica gneisses of the Kalak Nappe Complex, adjacent to a lens of preserved
Magerøya Nappe unit on the Porsanger Peninsula (Kirkland et al., 2007). This fault comprises thin,
dm-scale lenses of dark clay particles along the fault-core and damage zone with splaying fault
geometries (Figure 2h). Oblique normal-sinistral movement is inferred from slickenside lineations
(Figure 2h), and the amount of displacement probably does not exceed a few tens of meters because
garnet-bearing gneisses occur on both sides of the fault.

Farther north, sample 9 corresponds to a steep, WNW-ESE to E-W trending, S-dipping
fault (Figure 1 and Figure 2i) in a new quarry within a suite of weakly foliated gabbroic rocks of
the Honningsvåg Igneous Complex (Corfu et al., 2006). The sampled fault-core includes two thin,
5-10 cm-thick layers of light-colored, non-cohesive clay particles (Figure 2i).

The last fault we sampled for K/Ar dating (sample 10) was taken along a steep, WNW-ESE
striking, NNE-dipping brittle fault in the western part of Magerøya (cf. Figure 1 and Figure 2j).
This fault is part of a high-frequency, WNW-ESE trending lineament and brittle fault system that
corresponds to fault segments of the TKFZ, which pervasively truncate metasedimentary rocks of
the Kalak Nappe Complex in western Magerøya (Koehl et al., submitted) and displays a ca. 0.5 m-
thick fault-core made up with light-colored clay particles (Figure 2j).

### 4.2. Mineralogy and microtextural analysis of cohesive fault-rock

#### 4.2.1. Cohesive fault-rocks within the Alta-Kvænangen tectonic window

In order to describe and analyze cohesive brittle fault-rock characters and mineralogical
and textural changes during cataclasis, the host rock characters are used as frame. Cohesive fault-
rock was found along most exposed brittle faults crosscutting Precambrian volcano-sedimentary
rocks of the Alta-Kvænangen tectonic window (Figure 2a & b). The host rocks are fairly
undeformed but underwent low-grade (greenschist facies) metamorphic conditions during the
Svecofennian Orogeny , and developed a weak, bed-parallel foliation (Bøe & Gautier, 1978; Zwaan
& Gautier, 1980; Bergh & Torske, 1988). This foliation comprises partly recrystallized, sigma-



shaped grains of quartz and feldspar locally incorporated into an S-C foliation made of elongated
crystals of white mica (Figure 3a).

Brittle faults analyzed in the present study crosscut both meta-sandstones and
metamorphosed carbonate-rich host-rocks, and include three types of cataclasites and mineral
precipitations. The first type shows a matrix of finely crushed clasts of quartz (Figure 3b). The
second type is made of a partly healed, calcite-cemented cataclasite, which crystals display both
type II and IV twinning (Lerman 1999; Figure 3c). The third type includes cataclasite with
abundant brownish to reddish matrix of very fine-grained clay- (smectite and subsidiary illite) and
iron-rich minerals associated with iron-bearing precipitations, often truncating veins of
recrystallized quartz and quartz-rich cataclasite (Figure 3b). Iron-bearing precipitations often
appear to localize along fractures developed parallel to pre-existing, white-mica, S-C-C' foliation
(Figure 3b). Relative timing of quartz-, calcite- and clay-rich cataclasite could not be directly
resolved from crosscutting relationships.

### 4.2.2. Cohesive fault-rocks within Caledonian nappes

Metamorphic Caledonian host rocks consist of a variety of granodioritic gneisses,
metasandstones, metapelites, amphibolites/metavolcanics, gabbros and micaschists. When
truncated by brittle faults, most rocks are altered and/or display retrograde mineral assemblages.
Notably, in mafic/granodioritic host rocks, biotite is systematically retrograded into chlorite in the
vicinity of brittle faults (Figure 3d), host rocks are generally enriched in epidote at the expense of
amphibole, and garnet porphyroblasts are highly fractured (Figure 3e). Brittle faults crosscutting
Caledonian rocks comprise up to 1-2 m wide lenses of fractured host rocks showing preserved
ductile fabrics, such as widespread muscovite-biotite and S-C-C' foliation with feldspar sigma-
clasts partly recrystallized into quartz (e.g. along the Sørkjosen and Snøfjorden-Slatten faults;
Figure 3d & f). Typically, S, C and C' foliation surfaces are partly replaced by cataclastic fault-
rocks and, thus, may have localized subsequent brittle faulting (Figure 3g), as observed along the
brittle-ductile Talvik fault (Figure 2e).

We identified five texturally different types of cataclasites crosscutting each other in a
systematic order. The first type of cataclasite (type 1) is made of fine-grained, rounded to sub-
rounded clasts of epidote and chlorite (Figure 3e & h), often reworked into large, angular clasts
incorporated into subsequent cement or cataclastic matrix (Figure 3e & i). Epidote-chlorite fracture

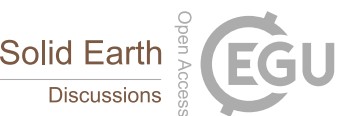

precipitations sometimes appear undeformed (Figure 3j). Occasionally, epidote-chlorite bearing
veins comprise rounded clasts of a brownish mineral showing moderate to strong relief (Figure
3h). EDS analysis reveals that this mineral is enriched in calcium, iron, silica, aluminium,
magnesium and oxygen (Figure 3k), and, thus, may correspond to stilpnomelane (Eggleton, 1972).

The second type of cataclasite (type 2) is composed of very fine-grained, rounded to sub-
rounded clasts of quartz (Figure 3h). This type of cataclasite is often observed adjacent to host rock
clasts (Figure 3e), regularly incorporates angular clasts of epidote-rich cataclasite (Figure 3e & h),
and occurs in conjunction with veins of recrystallized quartz (Figure 3h).

The third type of cataclasite (type 3) is widespread along fault segments and splay-faults of
the LVF (e.g. Sørkjosen, Straumfjordbotn, Langfjorden, Øksfjorden, Altafjorden 1 & 2, and Talvik
faults) and is characterized by poorly sorted, angular clasts of calcite (Figure 3l) often associated
with abundant, locally undeformed, calcite cement, showing type I and type II twinning (Figure 3i;
Lerman 1999). Calcite crystals consistently crosscut veins of recrystallized quartz, and epidote-
and quartz-rich cataclasites (Figure 3e).

The fourth type of cataclasite (type 4) consists of abundant, new-grown, mildly cataclased,
prismatic/columnar mineral grains with acute edges and steep-oblique terminations showing low
relief, low refraction index and three sets of cleavage (Figure 3e & m). We interpret this mineral
as laumontite, i.e. a high-temperature zeolite mineral (Dill et al., 2007; Triana et al., 2012). In the
study area, laumontite crystals commonly grew with their long-crystallographic axis perpendicular
to brittle fractures (Figure 3m) and consistently crosscut epidote- and quartz-rich cataclasites and
epidote-chlorite and calcite-filled veins (cf. Figure 3e).

The fifth type of cataclasite (type 5) shows enrichment in very fine-grained, iron-oxide
bearing mineral precipitations and an even more fine-grained, microscopic matrix of brownish and
greyish clay minerals (Figure 3e, h, l & n). These cataclastes truncate and often incorporate clasts
of epidote-chlorite, quartz- and calcite- rich cataclasite and associated mineral veins (cf. Figure 3e,
l & n).

### 4.3. Fault gouge mineralogy and K/Ar ages

*4.3.1. Mineralogy*





XRF analyses of various grainsize fractions of the sampled fault gouges in NW Finnmark
consistently show (1) high smectite content often associated with chlorite (mixed-layer chlorite-
smectite), e.g. samples 1, 2, 6 & 8 (Figure 1), (2) a relatively low content in illite and (3) variable
amount of residual quartz (Table 1 &Appendix A). The only exception is sample 8 from the
Porsanger Peninsula (cf. Figure 2h), which contains higher amounts of illite and quartz together

with smectite, chlorite and kaolinite clay minerals (Table 1 & Appendix A). The analysis of the
diffraction spectrum for all three grainsize fractions indicates that fault gouge from western
Magerøya (sample 10) is made of almost pure smectite (Figure 2j, Table 1 and Appendix A). Traces
of K-feldspar, indicated by minor peaks at 27.3-27.4 on inclined spectra (Table 1 & Appendix A),
were observed in all three grainsize fractions of fault gouge samples 1 & 2 in Sørkjosen (Figure 1,

Figure 2c & d, Table 1 & Appendix A), in the coarse (2-6 µm) fraction of sample 6 from the eastern
shore of Altafjorden (Figure 1, Figure 2f, Table 1 & Appendix A), in the coarse and intermediate
fractions of sample 7 from the Snøfjorden-Slatten fault on the Porsanger Peninsula (Figure 1,
Figure 2g, Table 1 & Appendix A), and in sample 10 along a WNW-ESE trending fault in western
Magerøya (Figure 1, Figure 2j, Table 1 & Appendix A). Sample 9 from the Honningsvåg Igneous

Complex on Magerøya (Figure 2i) shows very low potassium content (cf. Table 2), which resulted
in high 2σ errors associated with the K/Ar ages obtained for all three fractions (cf. Table 2). We
also noticed the presence of possible laumontite and/or stilbite in this sample (Table 1 & Appendix
A; Triana et al., 2012).

*4.3.2.  K/Ar dating results*

*Precambrian ages*

        All three dated fractions of sample 3 and 4 (Figure 1 & Figure 2a & b) yielded Precambrian
ages (Table 2, Figure 4 & Figure 5). For sample 3, the coarse fraction yielded a late

Mesoproterozoic age (1050.7 ± 12.2 Ma; cf. Figure 4 & Figure 5). The intermediate and finest
fractions of this sample both yielded early Cryogenian (Neoproterozoic) ages, 806.4 ± 10.7 and
824.7 ± 12.7 Ma respectively (Table 2 & Figure 4 & Figure 5). The intermediate fraction yielded
a slightly younger age in tendency compared to the finest fraction, taking the errors into account,
both ages do not differ significantly. Nevertheless, these K/Ar ages from well-preserved, non-

cohesive fault gouges suggest that the Altafjorden fault 1 (Figure 2a) formed in late





Mesoproterozoic times and was reactivated at least once in the Neoproterozoic (early Cryogenian). Younger, post-Caledonian reactivation seems unlikely, as the sensitivity of the K/Ar geochronometer would certainly have recorded subsequent reactivation by yielding younger ages.

Similar Precambrian K-Ar ages were obtained for the three fractions of sample 4 from the Altafjorden fault 2 (Figure 1, Figure 2b, Figure 4 & Figure 5). Here, the coarse-grained fraction yielded a late Mesoproterozoic age of 1054.2 ± 14.7 Ma, the intermediate fraction a Tonian (early Neoproterozoic) age of 943.8 ± 17.3 Ma, and the finest fraction a Cryogenian (mid Neoproterozoic) age of 811.3 ± 17.8 Ma (Table 2, Figure 4 & Figure 5). Considering the proximity of the Altafjorden faults 1 and 2 (samples 3 & 4; Figure 1) and the comparable ages obtained for these faults, we

consider the K/Ar ages to be reliable and likely reflecting protracted Mesoproterozoic-Neoproterozoic tectonic events. These results further suggest that the Altafjorden faults 1 and 2 were not reactivated in the Phanerozoic, as further reactivation would necessarily have been recorded and resulted in younger ages.

*Late Paleozoic ages*

        Most of the dated fault gouges from segments of the LVF in the Kalak Nappe Complex, as well as those along the TKFZ on the Porsanger Peninsula and Magerøya yielded late Paleozoic ages (Table 2, Figure 4 & Figure 6). The coarse fraction of the Talvik fault (sample 5; Figure 1 & Figure 2e) yielded a Silurian age of 427.3 ± 8.4 Ma, the intermediate fraction a Tournaisian (early

Carboniferous) age of 353.7 ± 4.1 Ma and the finest fraction an early Permian age of 282.1 ± 6.3 Ma (Figure 4 & Figure 6). These ages may indicate brittle faulting at the end of the Caledonian Orogeny and reactivation during post-Caledonian extension in the early Carboniferous-mid Permian, (see discussion).

        The intermediate and fine-grained fractions of sample 6 along the eastern shore of

Altafjorden (Figure 1 & Figure 2f) yielded earliest Permian and latest Carboniferous ages respectively (292.6 ± 4.0 Ma; 298.5 ± 5.0 Ma), while the coarse-grained fraction yielded a Jurassic age of 208.5 ± 3.1 Ma (Table 2, Figure 4 & Figure 6). A possible explanation for this discrepancy is that the coarse fraction of sample 6 partly consists of aggregates of smaller grains reflecting a much younger faulting event (Hamilton et al., 1989; Heizler & Harrison, 1991). However, XRF

analysis of the sample suggests that the anomalous, younger age obtained for the coarse fraction



may be the product of excess potassium due to the presence of K-feldspar in the sample (cf. Table 1 & Appendix A; Lovera et al., 1989).


The sample taken along the low-angle, WNW-ESE trending fault on the Porsanger Peninsula (sample 8; Figure 1 & Figure 2h) yielded similar latest Carboniferous-earliest Permian ages all included within a 5 Ma time span of $302.3 \pm 6.5$ Ma, $297.6 \pm 7.7$ Ma and $296.6 \pm 3.8$ Ma (cf. Table 2, Figure 4 & Figure 6). This suggests that the fault was not reactivated after the earliest Permian, as the finest fraction would have recorded a younger faulting event and, thus, yielded a younger age. It is, however, possible that the fault accommodated earlier faulting events.


Fault gouge sampled along a WNW-ESE trending fault segment of the TKFZ near Gjesvær in western Magerøya (sample 10; Figure 1 & Figure 2j) yielded late Carboniferous ($312.5 \pm 8.7$ Ma), early Permian ($284.0 \pm 6.2$ Ma) and mid Permian ($270.8 \pm 6.0$ Ma) ages, respectively, for the coarse, intermediate and finest fractions, suggesting the fault experienced multiple extensional faulting events from the late Carboniferous to mid-Permian (Table 2, Figure 4 & Figure 6).



Fault gouge sample 9 from an E-W to WNW-ESE trending fault within the Honningsvåg Igneous Complex in the Magerøy Nappe (Figure 1 & Figure 2i), contains very low amounts of potassium (Table 2). This, together with a high contamination of atmospheric argon, resulted in high errors for all three dated fractions. Considering the high 2σ error percentage associated to the K/Ar ages obtained for this sample, the coarse fraction may cover a time span of faulting from the mid-Carboniferous (early Serpukhovian) to the Late Pennsylvanian (Gzhelian), $315.6 \pm 13.6$ Ma (Table 2, Figure 4 & Figure 6). The finest fraction exhibits an even higher 2σ error percentage, and the age window included within 2σ interval spans from the early Permian to the Middle Triassic (Table 2, Figure 4 & Figure 6). The intermediate fraction yielded a younger age ($234.7 \pm 18.0$ Ma) than the finest fraction ($265.2 \pm 23.6$ Ma), which is considered to be erroneous. Brittle faulting along this fault most likely initiated in the mid-Carboniferous and the fault was later reactivated in the Permian.


*Mesozoic ages*


Mesozoic K-Ar ages were obtained for the Snøfjorden-Slatten fault on the Porsanger Peninsula (sample 7; Figure 1 & Figure 2g), which yielded Middle ($238.0 \pm 5.4$ Ma) - Late Triassic ($227.4 \pm 5.3$ Ma) to Hettangian (earliest Jurassic; $200.4 \pm 6.0$ Ma) ages (Table 2, Figure 4 & Figure 6). This sample, however, contains minor K-feldspar in the coarse and intermediate grainsize



fractions, which may have induced an excess of potassium, hence yielding younger ages than the age of faulting (Lovera et al., 1989). Nonetheless, the Hettangian age obtained for the finest fraction seems reasonable and most likely reflects an actual faulting event.

Fault gouge of the coarse, intermediate and fine grainsize fractions of sample 1 taken near the Sørkjosen fault segment of the LVF (Figure 1 & Figure 2c) yielded a latest Triassic (Rhaetian) age of 206.8 ± 2.6 Ma, and Late Jurassic (late Kimmeridgian) ages of 153.2 ± 3.7 Ma and 153.4 ± 1.9 Ma, respectively (Table 2, Figure 4 & Figure 6), possibly suggesting that this fault splay of the LVF formed in the latest Triassic and was reactivated in the Late Jurassic. Similar K-Ar ages were

obtained for another splay fault of the LVF in Sørkjosen (sample 2; Figure 1 & Figure 2d), i.e. Olenekian 247.6 ± 3.7 Ma and 249.4 ± 3.3 Ma (Early Triassic) ages for the coarse and intermediate fractions and a latest Mid Jurassic age of 164.4 ± 4.5 Ma for the finest fraction (Table 2, Figure 4 & Figure 6). Similarly to the other fault in Sørkjosen (sample 1), it is possible that the gouge in sample 2 formed in the Early Triassic and was reactivated during the Mid Jurassic (Table 2, Figure

4 & Figure 6). However, minor K-feldspar content observed in the diffraction spectrums of all three grainsize fractions of both of these faults suggest the K-Ar ages probably post-date the actual faulting, but it is uncertain by how much time (Table 1 & Appendix A).

## 5.  Discussion

        We combine mineral assemblages in cohesive and non-cohesive fault-rocks to reconstruct the faulting and burial-exhumation history of the NW Finnmark margin, using an average geothermal gradient of 30°C/km, and utilizing the K-Ar dating results of authigenic illites in non-cohesive fault-rocks to constrain the timing of faulting. The discussion starts with the estimated p-

T conditions and the Mesoproterozoic-Neoproterozoic K/Ar ages obtained for the Altafjorden faults 1 & 2, and proceeds with mid-late Paleozoic, and, finally, Mesozoic exhumation (p/T) and faulting data obtained from the LVF and TKFZ as basis for comparison with p-T constraints and K/Ar faulting ages from in Western Troms.

**5.1. Mesoproterozoic-Neoproterozoic faulting and exhumation history**

*5.1.1.   Evolution of temperature conditions in Precambrian rocks*



Microtextural and mineralogical analysis of cohesive fault-rocks along the Altafjorden faults 1 & 2 (Figure 2a & b) show that brittle faulting initiated with the formation of quartz- and calcite-rich cataclasites (Figure 3b & c). On the one hand, quartz-rich cataclasite derived from a foliated, meta-psammitic host rock with quartz/feldspar sigma-clasts (Figure 2a & b and Figure 3a). On the other hand, calcite-cemented cataclasite often incorporate crystals with type II and IV twinning, which indicate that these crystals were subjected to temperature ranges of 150-300°C (i.e. 5-10 km depth) and > 250°C (depth > 8 km) respectively (Figure 3c; Lerman, 1999).

Quartz-rich and calcite-cemented cataclasites are truncated and occasionally incorporated into subsequent iron/clay-rich cataclasites (Figure 3b & c). XRF analyses of non-cohesive fault-rocks sampled along the Altafjorden faults 1 & 2 (samples 3 & 4) show a dominance of smectite (Table 1 & Appendix A), which suggests that the dominant clay mineral in related iron/clay-rich cohesive fault-rock shown in Figure 3b is smectite. Considering such a predominance of authigenic smectite in both non-cohesive and cohesive, iron/clay-rich fault-rocks (Figure 3b, Table 1 & Appendix A), and assuming a complete diagenetic transformation of smectite into illite at ca. 105°C (cf. Morley et al., 2018) and a complete absence of authigenic illite at temperature < 35°C (Eberl et al., 1993), we propose that clay-rich fault-rocks along brittle faults in the Alta-Kvænangen tectonic window formed at temperature conditions comprised between 35-105°C (i.e. 1-3.5 km depth). Although crosscutting relationships of calcite-rich cataclasite with quartz- and iron/clay-rich fault-rocks are unknown, the irreversibility of the diagenetic transformation of smectite into illite (Eberl et al., 1993) suggests that calcite-cemented cataclasite, which formed at 5-10 km depth, is older than the iron/clay-rich (cohesive and non-cohesive) fault-rocks, which formed at shallow depth 1-3.5 km. Hence, we argue that Precambrian basement rocks in Altafjorden experienced at least three brittle faulting events, starting with quartz-rich cataclasite (Figure 3b) and/or calcite-cemented cataclasite formed at a depth of 5-10 km (Lerman, 1999; Figure 3c). Then, basement rocks were exhumed to shallow crustal level < 3.5 km (Morley et al., 2018) when the final, iron- and smectite-rich faulting event occurred (Figure 3b).

*5.1.2. Timing of faulting and exhumation of Precambrian rocks*

The latest Mesoproterozoic (ca. 1050 Ma) – early Neoproterozoic ages (ca. 945 Ma) obtained for the coarse fraction of sample 3 and the coarse and intermediate fractions of samples 4 (Table 2, Figure 4 & Figure 5), and the slickenside lineations and drag-folded foliation indicating



down-to-the-NNW normal motions along both faults (Figure 2a & b) suggest that the Altafjorden
faults 1 & 2 contributed to the initial stages of formation of the NW Baltoscandian basins
(Siedlecka et al., 2004; Nystuen et al., 2008) during the rifting of the Asgard Sea (Cawood et al.,
2010; Cawood & Pisarevsky, 2017). Possible driving mechanisms for the formation of these basins
and faults are a far-field influence of the coeval, basin-oblique/orthogonal, Sveconorwegian
contraction, i.e. a formation as impactogenic rift-basins (Barberi et al., 1982), and/or a possible
influence of late/post-orogenic collapse of the Sveconorwegian Orogeny (Bingen et al., 2008; Viola
et al., 2013). The latest Mesoproterozoic-early Neoproterozoic ages (ca. 1050-945 Ma) obtained
on illite in coarsest and intermediate fractions of brittle fault-rocks (Table 2, Figure 4 & Figure 5)
suggest that basement rocks of the Alta-Kvænangen tectonic window were already exhumed above
the brittle-ductile transition at that time, and may provide a maximum estimate for the age of quartz-
595 and calcite-rich cataclasites formed at depth of 5-10 km along these faults (Figure 3b & c).

The finest fractions of both samples and intermediate fraction of sample 3 of non-cohesive
fault-rocks in basement rocks yielded mid-Neoproterozoic ages (ca. 825-810 Ma; Table 2, Figure
4 & Figure 5). Combining these ages with normal shear-sense indicators observed in the field
(Figure 2a & b), we propose that they represent the onset of rifting of the Iapetus Ocean-Ægir Sea
during the breakup of Rodinia between 825 and 740 Ma (Torsvik & Rehnström, 2001; Hartz &
Torsvik, 2002; Li et al., 2008). Similar Neoproterozoic K/Ar ages of ca. 790-780 Ma and 740-735
Ma are reported from dating of authigenic illite/muscovite along the Kvenklubben and Porsavannet
faults in the adjacent Repparfjord-Komagfjord tectonic window in NW Finnmark (Torgersen et al.,
2014; Figure 1).

The Altafjorden faults 1 & 2, although located close and oriented sub-parallel to Caledonian
thrust faults (e.g. the Talvik fault; Figure 2e) and major, post-Caledonian normal faults (e.g. the
LVF; Figure 1), were most likely not reactivated after ca. 810 Ma (mid-Neoproterozoic; Table 2,
Figure 4 & Figure 5), as subsequent faulting would have triggered younger mineral assemblages
and ages. Possible explanations for the non-reactivation of these faults include a north/westwards
(basinwards?) migration of rifting to areas adjacent to the LVF, e.g. the Kvenklubben and
Porsavannet faults dated at ca. 790-735 Ma (Torgersen et al., 2014), and to faults in Troms) and
northern Finnmark, where Ediacaran metadolerite dykes intruded basement rocks during the
breakup of the Iapetus Ocean-Ægir Sea (Zwaan & van Roermund, 1980; Siedlecka et al., 2004;
Nasuti et al., 2015). The lack of reactivation of Altafjorden faults 1 & 2, predominance of





authigenic smectite clay mineral in non-cohesive fault-rock (samples 3 & 4 in Table 1 & Appendix
A) and the irreversibility of smectite-illite transformation (Eberl et al., 1993) suggest that
Precambrian rocks of the Alta-Kvænangen tectonic window were exhumed and have remained at
shallow depth < 3.5 km since the mid-Neoproterozoic (ca. 825 Ma; Table 2, Figure 4 & Figure 5).
This conclusion is supported by predominance and preservation of authigenic smectite in similar

non-cohesive fault-rocks in the Repparfjord-Komagfjord tectonic window (Torgersen et al., 2014).

Exhumation rates during latest Mesoproterozoic-early Neoproterozoic normal faulting are
unknown. However, exhumation rate from the early (ca. 945 Ma and 5-10 km depth) to mid-
Neoproterozoic (ca. 825 Ma and 1-3.5 km depth) were probably in the range of ca. 10-75 m per
Ma, i.e. comparable to what is expected from average continental erosion rates (10-100 m per Ma;

Schaller et al., 2002; Eppes & Keanini, 2017 – their figure 5). This therefore suggests a period of
tectonic quiescence between opening of the Asgard Sea (first two, quartz/calcite-rich faulting
events) and the onset of Iapetus rifting (final, smectite-rich faulting event).

## 5.2. Phanerozoic faulting and exhumation history


### 5.2.1. *Evolution of temperature conditions in Caledonian rocks*

We described five types of cohesive cataclastic fault-rocks in NW Finnmark based on
mineralogical and textural descriptions. First, epidote- and chlorite-rich, stilpnomelane-bearing

cataclasite (Figure 3e & h-j) formed by faulting of Caledonian mafic schists and gneisses
(amphibolites; Figure 3d & e; Ramsay et al., 1979; 1985; Gayer et al., 1985) ) and is consistently
truncated by and incorporated into the other four types of cataclasites (Figure 3e & h-j), suggesting
that epidote/chlorite-rich cataclasites correspond to the earliest stage of brittle faulting recorded by
Caledonian rocks. The epidote + chlorite + stilpnomelane ± biotite mineral assemblages present

both in the epidote/chlorite-rich cataclasites and adjacent host rocks, where biotite is almost
completely recrystallized into chlorite (Figure 3d), indicate lower greenschist facies conditions
during this faulting event, which constrain the minimum temperature during faulting to ca. 300°C
(i.e. 10 km depth). Further, rounded clasts of stilpnomelane in epidote-rich cataclastic veins
onshore Magerøya (Figure 3h & k) suggest faulting temperatures at prehnite-pumpellyite to lower

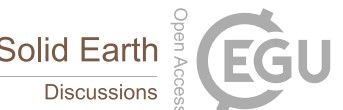

greenschist facies conditions comprised between 300°C and 470°C (Miyano & Klein, 1989), i.e. a
depth range of 10-16 km.

The second type of cataclasite corresponds to very fine-grained, quartz-rich cataclasite and
veins of recrystallized quartz that often truncate and incorporate clasts of (type 1) epidote- and
chlorite-rich cataclastic veins (Figure 3e & h). Since quartz dissolution only occurs at temperatures

> 90°C (Worley & Tester, 1995) and deforms plastically at temperature > 300°C (Scholz 1988;
Hirth & Tullis, 1989), we argue that the quartz-rich cataclasite and associated quartz veins were
formed during a discrete, second faulting event at depth comprised between 3 and 10 km. The
transition from early, deep (10-16 km), epidote/chlorite-rich faulting to subsequent, shallower (3-
10 km) quartz-rich cataclasis indicates that Caledonian rocks were partly exhumed between the

two faulting events.

The third type of cataclasite is made up of widespread calcite both as clasts, cement and
precipitations (Figure 3e & i). This type of cataclasite crosscuts epidote/chlorite- (type 1) and
quartz-rich cataclasites (type 2), hence suggesting calcite-rich cataclasite (type 3) formed during a
younger (reactivation) faulting event (Figure 3e & i). Since calcite crystals of the cataclasite display

characteristic twinning type I and II (Figure 3i), we inferred a temperature range of 150-200°C
(Lerman, 1999) and a depth of 5-7 km during this tentative, third faulting event.

A fourth type of cataclasite is present along fault segments of the LVF and TKFZ, showing
pervasive laumontite clasts and precipitations (Figure 3m), and consistently truncates greenschist
facies cataclasites (types 1, 2 & 3). Laumontite crystals are commonly undeformed and often

appear as elongated crystals with their long axis perpendicular to fracture boundaries (Figure 3m).
These observations suggest that laumontite formed as late growth along opening extensional cracks
or in tension veins, most likely at a later stage of faulting than minerals in the greenschist facies
cataclasites. Laumontite crystals themselves often appear mildly cataclased (Figure 3e, l & m), thus
indicating that faulting persisted after the growth of laumontite. The temperature stability range of

laumontite is 50-230°C (Jové & Hacker, 1997), which suggests that syn/post-laumontite faulting
occurred at a depth range of ca. 2-8 km, i.e. probably shallower than faulting event associated with
epidote/chlorite- (type 1), quartz- (type 2) and calcite-rich (type 3) cataclasites.

The fifth type of cataclasite is composed of iron oxide and clay minerals that crosscut all
other cataclasite types and vein minerals (Figure 3h, l & n). XRF analyses of related non-cohesive

fault-rock show a consistent dominance of authigenic smectite with subsidiary mixed-layer



chlorite-smectite and minor illite (Table 1 & Appendix A), suggesting the dominant clay mineral observed in the fifth type of cohesive cataclastic fault-rocks is smectite. Based on the preservation of abundant authigenic smectite (cf. Table 1 & Appendix A) and on the irreversibility of the smectite-illite diagenetic transformation (Eberl et al., 1993), we propose that the ultimate (fifth)

faulting event(s) in Caledonian rocks in NW Finnmark occurred at temperatures < 105°C (Morley et al., 2018), i.e. depth < 3.5 km, and that Caledonian rocks have remained at such shallow depth through Mesozoic-Cenozoic times.

Locally, XRF analysis of cohesive fault-rocks along WNW-ESE trending fault segments of the TKFZ in western Magerøya (sample 10; Figure 1) show almost pure authigenic smectite (Table

1 & Appendix A). This constrains temperature during faulting to a minimum of 35-65°C (ca. 1-2 km depth) at which small amounts of illite may form (Eberl et al., 1993; Huang et al. 1993; Morley et al., 2018). Shallow faulting is further supported by the presence of mixed-layer chlorite-smectite clays (in samples 1, 2, 6 & 8; Table 1 & Appendix A), which suggests that smectite (and mixed-layer chlorite-smectite) authigenic clays formed by retrograde diagenesis (i.e. exhumation) of

crushed chlorite during faulting (Warr & Cox, 2001; Nieto et al., 2005; Haines & van der Pluijm, 2012). Thus, we argue that Caledonian rocks along the LVF and TKFZ experienced another (late-stage) phase of uplift/faulting and exhumation from zeolite facies conditions (2-8 km) to diagenetic conditions (1-3.5 km). Alternatively, chlorite-smectite in non-cohesive fault-rocks along fault-segments of the LVF (e.g. samples 1 & 2; Figure 1, Table 1 & Appendix A) formed during the

shallowest phase of smectite-illite clay mineral reaction, while interlayered illite-smectite (cf. sample 8; Table 1 & Appendix A) formed during deeper phases of this reaction due to higher, normal faulting-related burial in the hanging-wall of the LVF (Whitney & Northrop, 1988). However, more samples are needed in the hanging-wall of the LVF to verify this hypothesis (only sample 8; Figure 1).


### 5.2.2.  *Timing of Phanerozoic faulting and exhumation of Caledonian rocks*

*Late Paleozoic inversion of brittle-ductile Caledonian thrusts*

The coarse fraction of the Talvik fault (sample 5; Figure 1), a south-verging Caledonian

thrust in rocks of the Kalak Nappe Complex in Altafjorden, yielded a mid/late Silurian age (427.3 ± 8.4 Ma) suggesting that brittle faulting along this fault initiated during the latest stages of the





Caledonian Orogeny (Table 2, Figure 4 & Figure 6). Movement along the Talvik fault started with top-to-the-south, ductile Caledonian thrusting as shown by quartz sigma-clasts and shear bands in mylonitic foliation, likely at a depth > 10 km (Scholz, 1988; Hirth & Tullis, 1989), and continued

with down-to-the-north, brittle, normal dip-slip faulting truncating ductile fabrics (Figure 2e). The earliest evidence of late/post-Caledonian, normal faulting in northern Norway are Early Devonian ages obtained for inverted shear zones in Vesterålen (Steltenpohl et al., 2011). This suggests that the mid/late Silurian faulting age/event recorded along the Talvik fault (coarse fraction) might represent a phase of top-to-the-south, brittle (-ductile?), Caledonian thrusting, rather than late/post-

Caledonian normal faulting (Figure 2e). Exhumation of the Talvik fault to brittle depth < 10 km in the mid/late Silurian was most likely due to combined thrusting and erosion. Alternatively, this Silurian age reflects input from inherited illite/muscovite component as shown by a small illite peak with epizonal KI (< 0.25) in the coarse fraction of this sample (Appendix A), suggesting a faulting event younger than the Silurian.

Top-to-the-south, Silurian, brittle thrusting along the Talvik fault was followed by successive early Carboniferous (Tournaisian), 353.7 ± 4.1 Ma and early Permian, 282.1 ± 6.3 Ma faulting events obtained from the intermediate and finest gouge fractions respectively (Table 2, Figure 4 & Figure 6). These events likely reflect post-Caledonian, down-to-the-N reactivation as a normal fault during the collapse of the Caledonides. Extensional reactivation is supported by

normal dip-slip slickensides along the Talvik fault truncating the initial ductile fabrics (Figure 2e). Further support appears from the early Permian inversion of an analog Caledonian thrust in the Repparfjord-Komagfjord tectonic window, the Kvenklubben fault (Torgersen et al., 2014), and from offshore seismic studies on the Finnmark Platform, where a major Caledonian thrust, the Sørøya-Ingøya shear zone, was inverted in the Mid/Late Devonian-early Carboniferous (Figure 1;

Koehl et al., 2018).

*Late Paleozoic normal faulting*

Our dating efforts of brittle segments of the LVF and TKFZ outlined above revealed numerous and consistent, late Paleozoic ages (Table 2, Figure 4 & Figure 6). Obtained K-Ar ages

cover a time span from early Carboniferous (Tournaisian; one age) for the Talvik fault, late Carboniferous (three ages) for brittle faults on the Porsanger Peninsula and Magerøya (samples 8, 9 & 10; Figure 4 & Figure 6), to early-mid Permian (eight ages for samples 5, 6, 8, 9 & 10; Figure

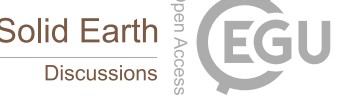

4 & Figure 6). By comparison, early (four ages) to late (one age) Carboniferous K-Ar ages were reported for the Markopp fault, and (two) early Permian ages for the Kvenklubben fault in nearby

rocks of the Repparfjord-Komagfjord tectonic window (Torgersen et al., 2014). In addition, the Laksvatn fault in Western Troms (Figure 1) yielded two Late Devonian ages (Davids et al. 2013). This down-to-the-NW normal fault is interpreted as a major, inverted Caledonian thrust possibly merging with the southwestern continuation of the LVF (Koehl et al., submitted), thus suggesting that post-Caledonian, normal brittle faulting along the LVF initiated in the Late Devonian. Further

support of Devonian faulting is found offshore, where potential Mid/Late Devonian sedimentary rocks were deposited along inverted Caledonian thrusts on the Finnmark Platform west, and where the offshore segments of the LVF on the Finnmark Platform east (Koehl et al., 2018) bound a major (half-) graben filled with Carboniferous (Bugge et al., 1995) and, conceivably, latest Devonian clastic sedimentary deposits (Roberts et al., 2011).

The dated faults in the Porsanger Peninsula (sample 8; Figure 2h), Talvik (sample 5; Figure 2e) and Storekorsnes (sample 6; Figure 2f) show down-to-the-north, normal dip-slip to oblique-slip movements, suggesting that they are all related to late Paleozoic, post-Caledonian extension. The long time-spread of the obtained late Paleozoic, post-Caledonian K/Ar ages (Table 2, Figure 4 & Figure 6) suggests either a long-term progressive, or two discrete faulting periods. From our

results, we favor two discrete periods, one in the Late Devonian-early Carboniferous at ca. 375-325 Ma (based on one age in this study, four from Torgersen et al., 2014 and two from Davids et al., 2013) and one in the late Carboniferous-mid Permian at ca. 315-265 Ma (eleven ages from the present study and two from Torgersen et al., 2014).

The obtained earliest Permian (Asselian) ages for three different fractions of the same

cataclasite in the fault in the hanging-wall of the LVF on the Porsanger Peninsula (sample 8, Figure 1, Table 2, Figure 4 & Figure 6), verified within two-sigma error range, may reflect a single faulting event. The short time span suggests that the fault was not reactivated later, since further faulting would have been recorded in the finest grainsize fraction. This conclusion is supported by offshore seismic data on the Finnmark Platform, showing that the thickness of Permian sedimentary rocks

is constant across brittle normal faults, such as the LVF and Måsøy Fault Complex (Figure 1), and that most brittle faults die out within the Carboniferous and lower part of the Permian sedimentary successions (Koehl et al., 2018). However, offshore seismic data on the Finnmark Platform suggest that early-mid Permian, K/Ar ages (Table 2, Figure 4 & Figure 6 and Torgersen et al., 2014)



obtained onshore NW Finnmark may represent only minor tectonic adjustments rather than major
faulting events (Koehl et al., 2018). Thus, an alternative interpretation for the dominance of
Permian ages is due to partial overprinting/resetting of authigenic illite from the main early (-late?)
Carboniferous faulting period.

        Of the five dated faults that yielded late Paleozoic, post-Caledonian ages (samples 5, 6, 8,
9 & 10; Table 2 & Figure 4), only two of them included cohesive fault-rock (samples 5 & 6; Figure
1). For those that only comprised non-cohesive gouge (samples 8, 9 & 10) with predominance of
authigenic smectite clay mineral and subsidiary authigenic illite (Table 1 & Appendix A), it seems
reasonable to conclude that faulting occurred at shallow depth between 1-3.5 km (Eberl et al., 1993;
Morley et al., 2018) in the late Carboniferous-mid Permian (Figure 4 & Figure 6). The other two
faults that yielded late Paleozoic ages (samples 5 & 6; Figure 1) and comprise both cohesive and
non-cohesive fault-rocks (Figure 2e & f), likely formed at deeper crustal levels and higher p/T
conditions. Non-cohesive fault-rocks along the Talvik fault (sample 5), consisting of authigenic
smectite with minor illite, yielded early Carboniferous (intermediate fraction) and early Permian
(finest fraction) ages (Table 2), while cohesive fault-rocks along this fault are characterized by both
quartz- and calcite-rich cataclasites (types 2 & 3). These data, backed by crosscutting relationships
between quartz-, calcite- and clay-rich cohesive fault-rocks (types 2, 3 & 5; Figure 3e), suggest
that quartz- and calcite-rich cataclasites (types 2 & 3) along the Talvik fault formed in the Late
Devonian (?) - early Carboniferous at a depth range of ca. 3-10 km (Scholz, 1988; Hirth & Tullis,
1989; Worley & Tester, 1995; Lerman, 1999). Later on, these cataclasites were overprinted by non-
cohesive, smectite-rich, cohesive (type 5) and non-cohesive fault-rocks generated at depth of 1-3.5
km in the early Permian (Figure 6; Eberl et al., 1993; Morley et al., 2018). These data are consistent
with a partial exhumation of the Talvik fault and nearby Caledonian host rocks from the early
Carboniferous to early Permian, with average exhumation rate along this fault varying from < 185
(Silurian-early Carboniferous) to < 125 m per Ma (early Carboniferous-early Permian).

        Similarly, for the fault in Storekorsnes, the intermediate and finest fractions of smectite-
dominated, fault-gouge sample 6 (Figure 1, Table 1 & Appendix A) yielded latest Carboniferous-
early Permian ages (Table 2 & Figure 4), and the fault comprises epidote- (type 1), quartz- (type
2), zeolite- (type 4) and smectite/chlorite-smectite rich (type 5) cohesive fault-rocks (Figure 3e).
The obtained K/Ar ages, kinematic, down-to-the-NW, normal dip-slip character (Figure 2f) and
stability field of smectite suggest that smectite- and chlorite/smectite-rich cohesive and non-




cohesive fault-rocks found along this fault formed during normal faulting in the latest
Carboniferous-early Permian at depth of 1-3.5 km (Eberl et al., 1993; Morley et al., 2018). Epidote-
rich and stilpnomelane-bearing (type 1), and quartz- (type 2) and zeolite-rich (type 4) cataclasites
are all crosscut by smectite-rich (cohesive and non-cohesive) fault-rocks (Figure 3e) and reflect
deeper faulting depths (ca. 2-10 km; Scholz, 1988; Hirth & Tullis, 1989; Miyano & Klein, 1989;
Worley & Tester, 1995; Jové & Hacker, 1997; Lerman, 1999). Thus, we propose that cataclasites
type 1, 2 and 4 formed much earlier, possibly in the Late Devonian-early Carboniferous as
suggested by K/Ar dating along the Laksvatn (Davids et al., 2013) and Talvik fault (Table 2 &
Figure 4), and were later exhumed and overprinted by late Carboniferous-early Permian, smectite-
rich faulting. Tentative driving mechanisms for exhumation may have been an interplay between
normal faulting, footwall uplift and continental erosion. This is supported by extensive normal
faulting offshore, in (Late Devonian?) Carboniferous rocks, and by the presence of a major, mid-
Carboniferous erosional unconformity of pre-Pennsylvanian rocks on the Finnmark Platform
(Larssen et al., 2002; Koehl et al., 2018).

        Of importance in restoring the exhumation history of NW Finnmark is that authigenic
smectite is particularly dominant in non-cohesive fault-rocks in the footwall and along fault
segments of the LVF, although often associated to high amounts of chlorite along fault segments
of the LVF (e.g. Talvik fault and Sørkjosen faults; see samples 1, 2 & 5 in Table 1 & Appendix A),
whereas interlayered illite-smectite dominates in non-cohesive fault-rocks in the hanging-wall of
the LVF, e.g. sample 8 (cf. Figure 1, Table 1 & Appendix A). A plausible interpretation is that km-
scale, post-Caledonian downthrow to the northwest along the LVF (partly) enhanced the
exhumation of brittle faults and Caledonian rocks in the footwall, while hanging-wall segments of
the LVF remained at deeper levels, producing interlayered illite-smectite (cf. sample 8; Table 1 &
Appendix A), a deeper end-member product of the smectite-illite reaction (Whitney & Northrop,
1988). This conclusion may be partly falsified by the illite-smectite rich composition of fault-
gouges along the Porsavannet and Markopp faults in the footwall of the LVF (Torgersen et al.,
2014). However, these faults yielded significantly older ages (respectively mid Neoproterozoic-
mid Paleozoic and early Carboniferous) than the late Carboniferous-early/mid Permian ages of
smectite/chlorite-smectite rich fault-rocks from our study (Table 2). It is therefore possible that the
authigenic, interlayered illite-smectite in fault-rocks along the Porsavannet and Markopp faults
reflect earlier, deeper faulting periods.



Furthermore, XRF analyses of fault gouge from a segment of the TKFZ in western Magerøya (sample 10; Figure 1 & Figure 2j), yielding late Carboniferous to mid-Permian K-Ar ages (cf. Table 2, Figure 4 & Figure 6), show almost pure authigenic smectite (cf. Table 1 & Appendix A). This suggests that Permian faulting occurred at very low temperature of 35-65°C

(Eberl et a., 1993; Huang et al., 1993; Morley et al. 2018), i.e. depth of ca. 1-2 km, and remained at shallow crustal levels until present, thus preventing transformation of smectite to illite through diagenesis. Late Paleozoic exhumation and shallow faulting in NW Finnmark and along other portions of the Barents Sea margin, e.g. Lofoten-Vesterålen, are supported by Apatite Fission Track data, indicating that outcropping rocks northern Norway have remained at relatively low

temperature < 120°C, i.e. depth < 4 km, since mid-Permian times (Hendriks et al., 2007).

Considering K/Ar ages obtained on non-cohesive fault-rocks, and mineral assemblages and crosscutting relationships of the five types of cohesive fault-rocks, we estimate exhumation rates from the Silurian (ca. 425 Ma and 10-16 km depth) to Late Devonian (ca. 375 Ma and minimum 5-10 km depth) to be < 220 m per Ma, i.e. analogous to estimate along the Talvik fault (< 185 m

per Ma). Similarly, exhumation rates through the Late Devonian (ca. 375 Ma and minimum 5-10 km depth) - early Carboniferous (ca. 325 Ma and 2-8 km depth) faulting period were < 160 m, and < 115 m per Ma through the ultimate faulting period from the end of the early Carboniferous (ca. 325 Ma and maximum 2-8 km depth) to the mid-Permian (ca. 265 Ma and 1-3.5 km depth), i.e. similar to exhumation rates obtained along the Talvik fault (< 125 m per Ma). Decreasing

exhumation rate from Silurian to mid-Permian times might indicate progressively milder faulting activity along the margin, with exhumation rates in the late Carboniferous-mid Permian being comparable/slightly higher than average continental erosion rates thought to be in the order of 10-100 m per Ma (Schaller et al., 2002; Eppes & Keanini, 2017 – their figure 5). Thus, we propose that exhumation in the Silurian-early Carboniferous was driven by a combination of continental

erosion, thrusting and, later on, normal faulting, while exhumation in the late Carboniferous-mid Permian was mostly due to continental erosion with limited contribution of normal faulting.

Alternatively, the predominance of Permian faulting ages obtained onshore NW Finnmark may be attributed to a period of extensive weathering in the (late Carboniferous- ?) early-mid Permian (cf. Table 2, Figure 4 & Figure 6), possibly reflected by highly weathered host rocks and

brittle fault surfaces showing no kinematic indicators onshore Magerøya (sample 9 & 10; Figure 1 and Figure 2i & j). Although this weathering may be related to much younger processes (Olesen et



al., 2012; 2013), Carboniferous-Permian, (sub-) tropical climate conditions prevailed in Baltica (Stemmerik, 2000; Larssen et al., 2002; Samuelsberg et al., 2003) and, hence, may have initiated weathering of exposed, uplifted footwall blocks along major faults like the LVF and Måsøy and

Troms-Finnmark fault complexes offshore (Figure 1). This is supported by widespread exhumation and erosional truncation of pre-Pennsylvanian rocks in the footwall of the Troms-Finnmark Fault Complex, on the Finnmark Platform (Koehl et al., 2018), linked to a mid-Carboniferous phase of eustatic sea-level fall (Saunders & Ramsbottom, 1986). This exhumation/weathering event is also consistent with the Early Mesozoic, minimum age estimate of weathering of basement rocks along

the Norwegian continental shelf (Olesen et al., 2012; 2013). Although the dated faults were still buried to depth > 1 km in the mid-Permian, as shown by the presence of (minor) authigenic illite (Eberl et al., 1993) used for K/Ar age dating of the faults, field studies in onshore tunnels in Norway show that weathering processes related to percolation of acidic water may penetrate the bedrock > 200 m along fault surfaces (Olesen et al., 2012; 2013), thus making this alternative explanation

possible, though unlikely. Another obstacle to this interpretation is the lack of a major erosional unconformity/truncation in upper Carboniferous-lower/mid Permian sedimentary rocks on the Finnmark Platform offshore (Larssen et al., 2002; Samuelsberg et al., 2003; Koehl et al., 2018).

*Mesozoic faulting*

Reliable Mesozoic K/Ar age was only obtained for the finest fraction of the Snøfjorden-Slatten fault (sample 7; Figure 1 & Table 2), yielding a Hettangian age. All the other faults yielding Mesozoic ages (all three fractions in samples 1 & 2, coarse fraction of sample 6, coarse and intermediate fractions of sample 7 and intermediate fraction of sample 9) comprise subsidiary K-feldspar (





Table 1 & Appendix A), which provided additional potassium and, thus, yielded younger ages than the actual age of faulting (Table 2; cf. red ages in Figure 4). Thus, we disregard these ages because of their high uncertainty. Considering the scarsity of Mesozoic-Cenozoic ages, we argue that NW Finnmark, as well as adjacent offshore areas of the Finnmark Platform (Koehl et al., 2018) were tectonically quiet after late Paleozoic (Devonian-mid Permian) extension and were

only subjected to minor, local extensional faulting events, e.g. in the earliest Jurassic (Hettangian) for the Snøfjorden-Slatten fault (cf. Figure 1, Table 2, Figure 4 & Figure 6) and Early Cretaceous for the Kvenklubben fault (Torgersen et al., 2014).

### 5.3. Regional implications


An implication of the latest Mesoproterozoic-Neoproterozoic K/Ar ages obtained for the ENE-WSW trending Altafjorden faults 1 & 2 in the Alta-Kvænangen tectonic window is that they partly support the interpretation of Koehl et al. (submitted), suggesting that ENE-WSW trending faults represent inherited Precambrian fault fabrics. However, the inferred normal sense of shear

and latest Mesoproterozoic-mid Neoproterozoic K/Ar faulting ages obtained for the Altafjorden faults 1 & 2 (Figure 2a & b) suggest that these faults formed as extensional normal faults rather than conjugate strike-slip faults to WNW-ESE trending faults like the TKFZ as suggested by Koehl et al. (submitted). Instead, latest Mesoproterozoic-mid Neoproterozoic brittle faults might have provided preferentially oriented weakness zones for the formation of subparallel, subsequent and adjacent Caledonian thrusts (e.g. Talvik fault) and post-Caledonian normal faults (e.g. LVF; Figure

1). Nevertheless, conjugate strike-slip faults may exist in NW Finnmark (Roberts, 1971; Worthing, 1984), but these display subvertical geometries and significant lateral displacement, and may have formed during E-W/ENE-WSW directed, Timanian contraction in the late Neoproterozoic, e.g. TKFZ (Siedlecka et al., 2004; Herrevold et al., 2009) and Akkarfjord fault (Roberts, 1971; Koehl

et al., submitted).

Analogous studies of post-Caledonian brittle faults in Western Troms show that post-Caledonian extensional faulting initiated at depth > 10 km at greenschist facies conditions and continued under pumpellyite-prehnite facies conditions at depth < 8.5 km, thus supporting a gradual exhumation of the margin (Indrevær et al., 2013, 2014). More detailed mineralogic-textural

analysis of clay-rich non-cohesive fault-rocks of the Vannareid-Burøysund, Sifjord and Laksvatn




faults revealed dominance of smectite and chlorite clay minerals (Davids et al. 2013), suggesting that brittle faults in Western Troms were exhumed to low temperature conditions (35-105°C; Eberl et al., 1993; Morley et al., 2018) and shallow depths (1-3.5 km) comparable the LVF and TKFZ in NW Finnmark. Furthermore, fault-gouge along the SSE-dipping Vannareid-Burøysund and Sifjord

faults yielded similar early Carboniferous (intermediate fractions) and early Permian K/Ar ages (finest fractions; Davids et al., 2013) compatible with the proposed Late Devonian-early Carboniferous and late Carboniferous-mid Permian stages of post-Caledonian brittle faulting in NW Finnmark (Table 2, Figure 4, Figure 6 and Torgersen et al., 2014).

A major contrast in K/Ar ages in Western Troms and NW Finnmark is occurrence of latest

Mesoproterozoic-mid Neoproterozoic ages for gouges of the southeasternmost normal faults within basement rocks of the Alta-Kvænangen tectonic window in NW Finnmark (Figure 4 & Figure 5), while analogous faults in Archean-Paleoproterozoic rocks of the West Troms Basement Complex (Zwaan, 1995; Bergh et al., 2010) yielded Carboniferous-Permian ages (Davids et al., 2013). Another mild contrast is the occurrence of slightly younger, late Permian-Early Triassic, K/Ar

faulting ages for brittle faults in Western Troms (Davids et al., submitted), suggesting that extension migrated westwards after the late Carboniferous-mid Permian and persisted until the Early Triassic in coastal areas of Western Troms. Westwards younging of K/Ar faulting ages is further supported by Mesozoic ages obtained for three faults in western Lofoten (Davids et al., 2013). Nonetheless, widespread Late Devonian-early Carboniferous and late Carboniferous-mid

Permian ages in NW Finnmark, Western Troms and Lofoten-Vesterålen suggest that the main episode of extension and exhumation along the margin occurred in the late Paleozoic and was probably related to the collapse of the Caledonides (Davids et al., 2013, submitted; Torgersen et al., 2014; Koehl et al., 2018). Apatite Fission Track data in Western Troms and Lofoten-Vesterålen also indicate a period of rapid cooling (1-2°C/Ma) in the late Paleozoic, possibly due to combined

extensive normal faulting and erosion, followed by a period of relatively slow cooling (< 0.2°C/Ma) in Mesozoic times, likely suggesting a tectonically quiet time period (Davids et al., 2013).

## 6.  Conclusions






1) Three faulting events occurred in the latest Mesoproterozoic-mid Neoproterozoic (ca. 1050-810 Ma), including (i) latest Mesoproterozoic faulting (ca. 1050 Ma.) and (ii) an early Neoproterozoic faulting event (ca. 945 Ma) with quartz-rich and calcite-cemented cataclasites formed at depth of ca. 5-10 km, possibly reflecting the formation of the NW Baltoscandian basins during the opening

of the Asgard Sea, and (iii) a shallow (depth 1-3.5 km), mid-Neoproterozoic faulting episode (ca. 825-810 Ma) with abundant authigenic smectite, related to the opening of the Iapetus Ocean-Ægir Sea and breakup of Rodinia between 825-740 Ma.

2) Exhumation rates estimates from 945 to 825 Ma were in the order of 10-75 m per Ma, thus indicating that continental erosion alone may account for early-mid Neoproterozoic exhumation

and that tectonic quiescence prevailed between the opening of the Asgard Sea and the opening of the Iapetus Ocean-Ægir Sea.

3) The preservation of abundant authigenic smectite in cohesive and non-cohesive fault-rocks suggests that Paleoproterozoic basement rocks were exhumed to and remained at shallow crustal levels (< 3.5 km depth) since the mid-Neoproterozoic (ca. 825 Ma), and were not reactivated after

mid-Neoproterozoic times despite being oriented parallel to major Caledonian thrusts and post-Caledonian normal faults.

4) Five faulting events occurred in Caledonian rocks, defining three faulting periods: (i) potential Silurian, top-to-the-south thrusting along Caledonian thrusts (e.g. Talvik fault) initiated at a depth of 10-16 km, and was possibly associated with epidote/chlorite-rich, stilpnomelane-bearing

cataclasis (type 1), (ii) widespread, Late Devonian-early Carboniferous (ca. 375-325 Ma) extensional faulting, occurred at decreasing depth and was accompanied by quartz-rich (type 2; 3-10 km depth), calcite-cemented (type 3; 5-7 km depth) and laumontite-rich cataclasites (type 4; 2-8 km depth) formed during three discrete faulting events possibly related to the collapse of the Caledonides, (iii) an ultimate, minor stage of shallow faulting in the late Carboniferous-mid

Permian (ca. 315-265 Ma) dominated by iron/smectite/chlorite-smectite rich, illite-bearing (type 5) fault-rocks formed at depth of 1-3.5 km, thus suggesting Caledonian rocks were progressively exhumed to near-surface depth in late Paleozoic times.

5) Km-scale, down-to-the-NW normal faulting and footwall uplift along the Langfjord-Vargsund fault may be responsible for local variation of dominant, authigenic clay minerals in type 5 fault-



rocks (1-3.5 km depth), producing deeper, interlayered illite-smectite in the hanging-wall and shallower, smectite and mixed-layer chlorite-smectite in the footwall.

6) Decreasing exhumation rates, < 220 m per Ma in Silurian-Late Devonian (425-375 Ma), < 160 m per Ma in Late Devonian-early Carboniferous (375-325 Ma) and < 115 m per Ma from mid-Carboniferous to mid-Permian times (325-265 Ma), suggest a transition from extensive,

widespread Caledonian thrusting and collapse-related normal faulting to milder normal faulting in the late Carboniferous-mid Permian. The high number of early-mid Permian, K/Ar ages may, alternatively, reflect an episode of (near-) surface weathering in NW Finnmark. Subsequent Mesozoic-Cenozoic extension migrated westwards and NW Finnmark remained tectonically quiet from the mid-Permian.


**Data availability**

Structural field measurements, analyzed thin sections and K/Ar geochronological data may be obtained from the corresponding author.

**Author contribution**

Jean-Baptiste Koehl acquired field measurements and fault-rock samples with the help of Prof. Steffen Bergh. K/Ar analyses were performed by Prof- Klaus Wemmer at the University of Göttingen and interpreted by Jean-Baptiste Koehl and Prof. Wemmer. The writing part was mostly done by Jean-Baptiste Koehl with the help of Prof. Bergh. Contributions as follow: Jean-Baptiste

Koehl (40 %), Prof. Steffen Bergh (30 %) and Prof. Klaus Wemmer (30 %).

**Competing interests**

The authors declare that they have no conflict of interest.

**Acknowledgements**

The present study is part of the ARCEx project (Research Centre for Arctic Petroleum Exploration), which is funded by the Research Council of Norway (grant number 228107) together with ten academic and eight industry partners. We would like to thank all the persons from these




institutions that are involved in this project. We acknowledge the contribution of student research assistants from the K/Ar laboratory at the University of Göttingen for their work preparing and analyzing the samples of non-cohesive fault-rock presented in this study. Finally, the authors would like to thank Anna Ksienzyk from the University of Bergen for fruitful discussions.

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





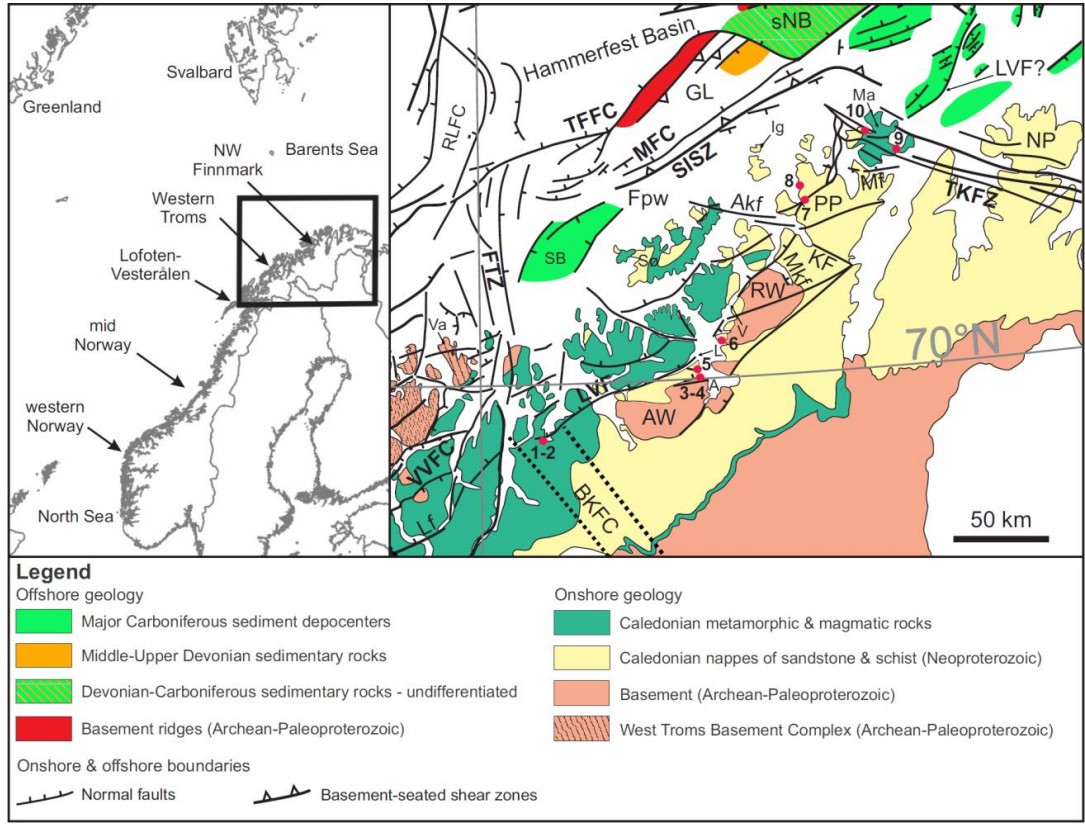

**Figure 1: The upper left inset shows the location of the study area (NW Finnmark) along the Norwegian continental shelf as a black frame. The upper right inset shows a tectonic map of NW Finnmark showing the location of dated brittle faults as red dots numbered as follows: (1) = Sørkjosen 1; (2) = Sørkjosen 2; (3) = Altafjorden 1; (4) = Altafjorden 2; (5) = Talvik; (6) = Storekorsnes; (7) = Snøfjorden-Slatten; (8) = Porsanger; (9) = Honningsvåg; (10) = Gjesvær. Map modified after Indrevær et al., (2013) and Koehl et al. (2018). Abbreviations: A = Altafjorden; Akf =Akkarfjord fault; AW = Alta-Kvænangen tectonic window; BKFC = Bothnian-Kvænangen Fault Complex; FPw = Finnmark Platform west; FTZ = Fugløya transfer zone; GL = Gjesvær Low; Ig = Ingøya; KF = Kokelv Fault; L = Langfjorden; LVF = Langfjord-Vargsund fault; Ma = Magerøya; Mf = Magerøysundet fault; MFC = Måsøy Fault Complex; Mkf = Markopp fault; NP = Nordkinn Peninsula; PP = Porsanger Peninsula; RLFC = Ringvassøya-Loppa Fault Complex; RW = Repparfjord-Komagfjord tectonic window; SB = Sørvær Basin; SISZ = Sørøya-Ingøya shear zone; sNB = southwesternmost Nordkapp basin; Sø = Sørøya; TFFC = Troms-Finnmark Fault Complex; TKFZ = Trollfjorden-Komagelva Fault Zone; V = Vargsundet; Va = Vanna; VVFC = Vestfjorden-Vanna fault complex.**





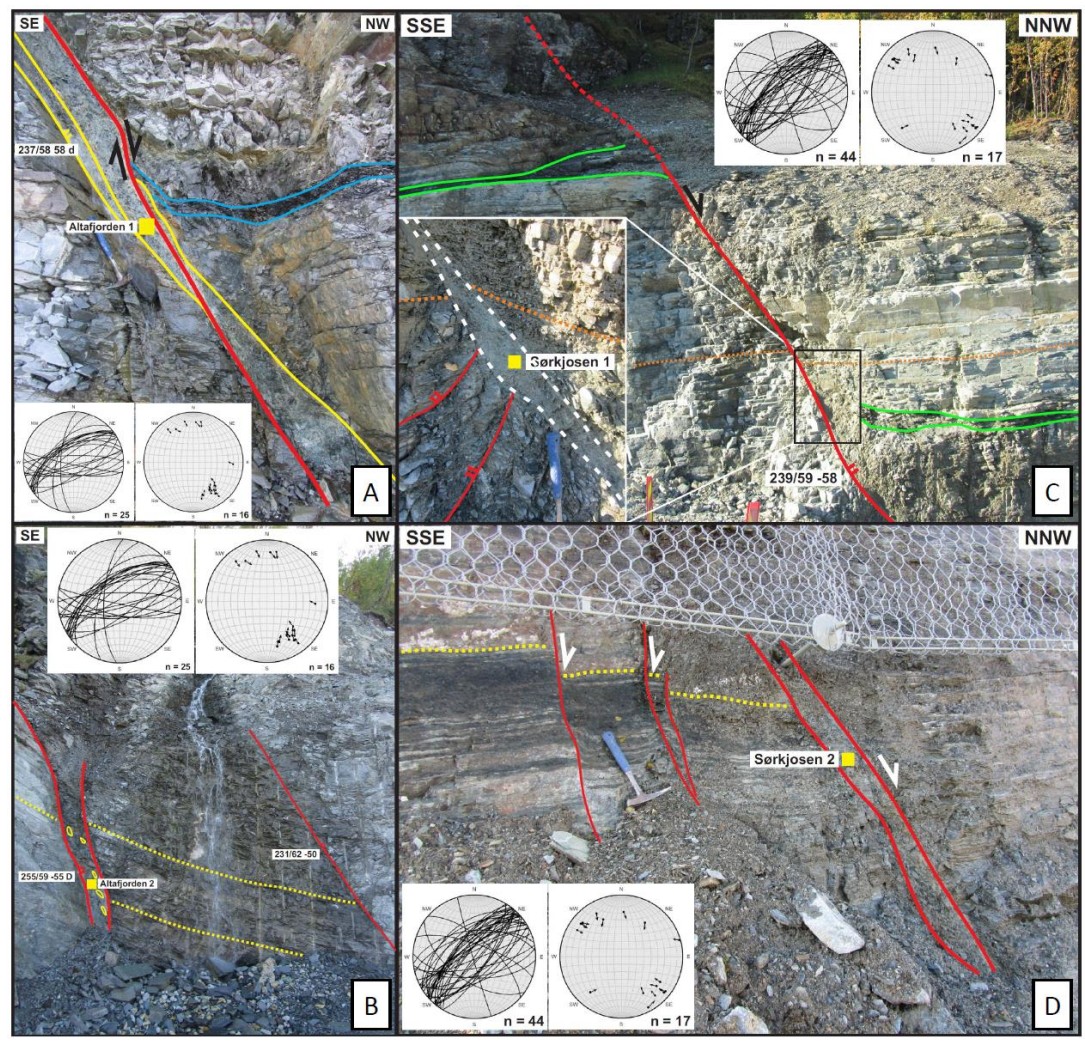



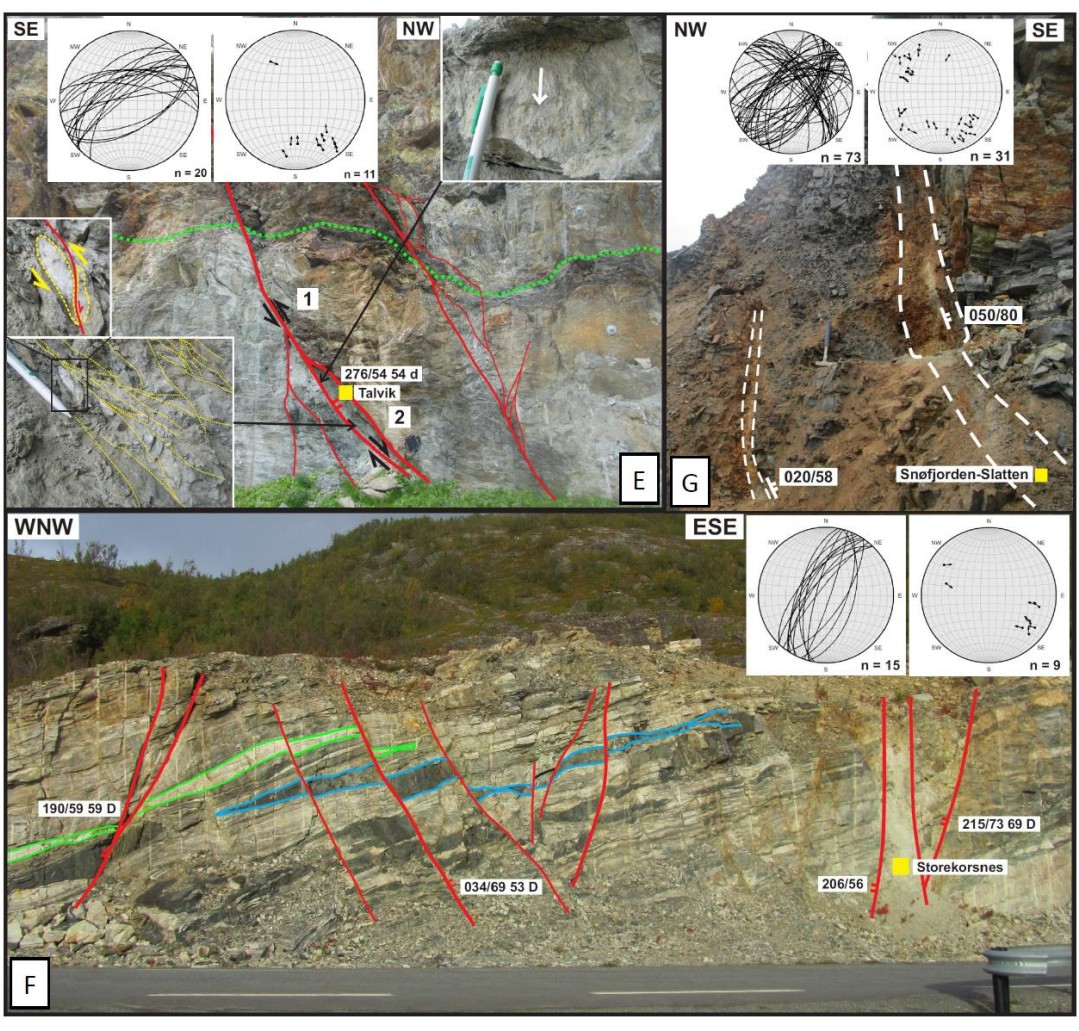

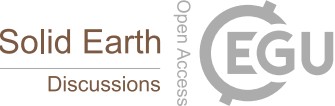





**Figure 2: Outcrop photographs of the dated faults. Each photograph is accompanied of the location of the dated samples (yellow squares), structural measurements presented in white boxes and in Schmidt stereonets (left hand-side stereonets show fracture surfaces as great circles and right hand-side stereonets display slickenside lineations as pole to fault surfaces indicating the movement of the hanging-wall), and potential kinematic indicators where available. (a) ENE-WSW trending, NW-dipping Altafjorden fault 1 (sample 3) crosscutting Precambrian rocks of the Alta-Kvænangen tectonic window along the western shore of Altafjorden. A potentially drag-folded mafic bed is shown in blue. Modified after Koehl et al (submitted); (b) ENE-WSW trending, NW-dipping Altafjorden fault 2 (sample 4) within the Alta-Kvænangen tectonic window. The country-rock displays preserved bedding surfaces (in dotted yellow); (c) ENE-WSW trending, NW-dipping fault in the footwall of the Sørkjosen fault (Sørkjosen 1; sample 1). A potentially offset mafic bed is shown in green and the bedrock fabric in dotted orange. The lower left frame is a zoom in the fault-core displaying the location of the dated sample; (d) ENE-WSW trending, NW-dipping fault in the footwall of the Sørkjosen fault (Sørkjosen 2; sample 2). Dotted yellow lines show normal offsets of mafic beds across brittle faults; (e) E-W trending, N-dipping Talvik fault (sample 5) crosscutting rocks of the Kalak Nappe Complex along the western shore of Altafjorden. Kinematic indicators include ductile fabrics made of microshears (lower left frame), sigma-clasts (middle left frame) and slickenside lineations (upper right frame). The bedrock fabric is shown in green. Modified after Koehl et al (submitted); (f) NNE-SSW trending brittle faults crosscutting rocks of the Kalak Nappe Complex along the eastern shore of Altafjorden (sample 6). See normal offsets of geological markers (in green and blue). Modified after Koehl et al (submitted); (g) NE-SW trending, SE-dipping fault segment of the Snøfjorden-Slatten fault (sample 7) in Snøfjorden, on the Porsanger Peninsula. Modified after Koehl et al (submitted); (h) Low-angle, WNW-ESE trending, NE-dipping fault (sample 8) crosscutting rocks of the Kalak Nappe Complex on the Porsanger Peninsula; (i) Steep, E-W to WNW-ESE trending faults (sample 9) within the gabbroic rocks of the Honningsvåg Igneous Complex; (j) Steep, WNW-ESE trending brittle faults near Gjesvær (sample 10), in the western part of Magerøya. The faults crosscus rocks of the Kalak Nappe Complex.**











**Figure 3: Microscope photographs of cataclasite and hostrock in NW Finnmark. (a)** Precambrian S-C foliation made of muscovite (ms) microcrystals surrounding K-feldspar (Kfs) sigma-clasts; **(b)** Cataclasite vein (dashed green) in Precambrian basement rocks seemingly offset by iron- and clay-rich fractures that formed parallel to existing S-C foliation deformation planes (dotted pink). The amount of offset across iron-rich fractures is shown by a dextrally (red) offset, sheared quartz grain (yellow) and is lower than the apparent offset of the cataclastic vein in green; **(c)** Calcite-filled (cal) fracture in Precambrian basement rocks crosscutting a vein of recrystallized quartz (qz) and a muscovite-bearing (ms) ductile microshear. Calcite cement crystals typically show type II (tw II; lower right inset) and type IV (tw IV; upper left inset) twinnings; **(d)** Preserved biotite (bt) foliation in Caledonian host rocks near a cataclastic (cata) brittle fault. Approaching the fault, biotite is increasingly recrystallized into chlorite (chl); **(e)** Highly fractured garnet crystal (grt) in Caledonian host rock crosscut by epidote- (ep; type 1) and quartz-rich cataclasites (qz cata; type 2), which are both truncated by calcite-filled veins (cal; type 3). Both cataclasites and calcite veins are truncated by a third cataclasite made up with a matrix of angular, poorly sorted clasts of laumontite (lmt; type 4) crosscut by late clay and iron-bearing veins (type 5); **(f)** Caledonian S-C foliation made of muscovite (ms) microcrystals surrounding K-feldspar (Kfs) sigma-clasts that have partly recrystallized into quartz (qz); **(g)** Cataclastic fault-rock showing the remains of pre-existing S, C and C' foliation

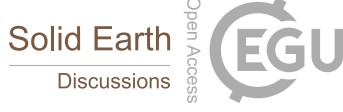



deformation planes in dotted yellow along which brittle fractures formed; (h) Epidote-rich cataclasite (ep cata; type 1) in Caledonian host rock including clasts of stilpnomelane (stp) crosscutting quartz-rich host rock (qz) and truncated by quartz-rich cataclasite (qz cata; type 2), which incorporates clasts of epidte-rich cataclasite (yellow; type 1). Ep- (type 1) and quartz-rich (type 2) cataclasites are truncated by subsequent clay and iron-rich veins (type 5); (i) Epidote-rich cataclasite (ep cata; type 1) embedded within a calcitic cement (cal; type 3) made of large crystals showing type I (tw I) and type II (tw II) twinnings; (j) Fracture with chlorite (chl) precipitation related with type 1 cataclasite; (k) SEM analysis of the atomic composition of the stilpnomelane crystal shown in type 1 cataclasite in (h). The numbers below the graph represent mass percentage of each atom; (l) Large clasts of laumontite-rich (lmt; type 4) cataclasite crosscut at the bottom of the photograph by a cataclastic vein including clasts of epidote (ep; type 1), quartz (type 2), calcite (cal; type 3) and laumontite (type 4), and on the right hand-side by a cataclastic vein containing mostly iron-bearing and clay minerals (type 5). The iron- and clay-rich cataclasite (type 5) truncates all the other types of cataclasites; (m) Laumontite precipitations (lmt; type 4 cataclasite). Crystals are elongated perpendicular to the fracture along which they precipitated; (n) Iron- and clay-rich (type 5) cataclasite crosscutting mildly fractured late growth of laumontite (lmt; type 4).





**Table 1: Mineral composition of dated fault gouges. "Presumed traces" are based on inclined spectra (cf. Appendix A) showing peaks at 27.3-27.4 for K-feldspar, and on the presence of minor illite in smectite, enabling K/Ar dating, e.g. samples 9 and 10. Abbreviations: chl = chlorite; I = illite; kao = kaolinite; Kfs = K-feldspar; lmt = laumontite; Qz = quartz; S = smectite.**

| Sample | Grainsize | S | I | Qz | Kfs | plagioclase | chl | kao | lmt? |
|---|---|---|---|---|---|---|---|---|---|
| 1 | <0.2μm | * | O | | - | | ** | | |
| | <2μm | * | O | | - | | ** | | |
| | 2-6μm | ** | O | * | - | | ** | | |
| 2 | <0.2μm | * | O | | - | | ** | | |
| | <2μm | * | O | | - | | ** | | |
| | 2-6μm | * | * | O | - | | ** | | |
| 3 | <0.2μm | ** | * | O | | | * | | |
| | <2μm | ** | * | O | | | * | | |
| | 2-6μm | ** | ** | * | | | ** | | |
| 4 | <0.2μm | ** | * | | | | O | | |
| | <2μm | ** | * | | | | * | | |
| | 2-6μm | ** | * | O | | | O | | |
| 5 | <0.2μm | ** | * | | | | * | * | |
| | <2μm | ** | * | O | | | ** | ** | |
| | 2-6μm | * | * | O | | | ** | ** | |
| 6 | <0.2μm | ** | * | | | | * | | |
| | <2μm | ** | * | O | | | * | | |
| | 2-6μm | ** | O | O | - | | * | * | |
| 7 | <0.2μm | ** | O | O | | | | | |
| | <2μm | ** | * | O | - | | | | |
| | 2-6μm | ** | * | * | O | | | | |
| 8 | <0.2μm | ** | * | | | | * | | |
| | <2μm | ** | ** | | | | * | | |
| | 2-6μm | ** | ** | O | | | * | | |
| 9 | <0.2μm | ** | - | | | | | | ** |
| | <2μm | ** | - | | - | | | | ** |
| | 2-6μm | ** | - | | | | | | ** |
| 10 | <0.2μm | ** | - | O | - | | | | |
| | <2μm | ** | - | O | - | | | | |
| | 2-6μm | ** | - | * | O | O | | | |

** major component     * minor component     O trace     - presumed trace

all smectites are of R=0 type, indicating maximum 10% of Illite content



**Table 2: K/Ar ages from synkinematic illite in fault gouge.**

| Sample | Spike [ No. ] | K2O [ Wt. % ] | 40 Ar * [ nl/g ] STP | 40 Ar * [ % ] | Age [ Ma ] | 2s-Error [ Ma ] | 2s-Error [%] |
|---|---|---|---|---|---|---|---|
| 1 | 5450 | 0,74 | 3,85 | 85,60 | 153,4 | 1,9 | 1,2 |
|   | 5467 | 1,07 | 5,52 | 90,43 | 153,2 | 3,7 | 2,4 |
|   | 5455 | 1,63 | 11,52 | 97,06 | 206,8 | 2,6 | 1,3 |
| 2 | 5461 | 0,70 | 3,89 | 43,30 | 164,4 | 4,5 | 2,7 |
|   | 5456 | 1,00 | 8,63 | 99,93 | 249,4 | 3,3 | 1,3 |
|   | 5457 | 2,29 | 19,57 | 96,08 | 247,6 | 3,7 | 1,5 |
| 3 | 5431 | 2,76 | 92,99 | 98,78 | 824,7 | 12,7 | 1,5 |
|   | 5437 | 3,11 | 102,07 | 98,46 | 806,4 | 10,7 | 1,3 |
|   | 5430 | 6,04 | 277,63 | 99,55 | 1050,7 | 12,2 | 1,2 |
| 4 | 5440 | 1,01 | 33,50 | 94,56 | 811,3 | 17,8 | 2,2 |
|   | 5442 | 2,77 | 110,73 | 98,23 | 943,8 | 17,3 | 1,8 |
|   | 5446 | 5,58 | 257,47 | 99,24 | 1054,2 | 14,7 | 1,4 |
| 5 | 5451 | 1,23 | 12,14 | 60,55 | 282,1 | 6,3 | 2,2 |
|   | 5428 | 1,52 | 19,11 | 92,83 | 353,7 | 4,1 | 1,2 |
|   | 5429 | 1,53 | 23,84 | 96,46 | 427,3 | 8,4 | 2,0 |
| 6 | 5435 | 1,34 | 14,04 | 65,30 | 298,5 | 5,0 | 1,7 |
|   | 5443 | 1,63 | 16,73 | 73,27 | 292,6 | 4,0 | 1,4 |
|   | 5434 | 1,25 | 8,90 | 67,82 | 208,5 | 3,1 | 1,5 |
| 7 | 5438 | 2,19 | 14,98 | 81,40 | 200,4 | 6,0 | 2,3 |
|   | 5454 | 3,79 | 29,66 | 89,43 | 227,4 | 5,3 | 3,0 |
|   | 5449 | 9,61 | 78,88 | 98,61 | 238,0 | 5,4 | 2,3 |
| 8 | 5465 | 3,64 | 37,86 | 92,47 | 296,6 | 3,8 | 1,3 |
|   | 5433 | 3,91 | 40,80 | 93,87 | 297,6 | 7,7 | 2,6 |
|   | 5432 | 5,17 | 54,91 | 98,19 | 302,3 | 6,5 | 2,2 |
| 9 | 5436 | 0,12 | 1,09 | 23,00 | 265,2 | 23,6 | 8,9 |
|   | 5448 | 0,19 | 1,50 | 26,04 | 234,7 | 18,0 | 7,7 |
|   | 5444 | 0,37 | 4,16 | 47,29 | 315,6 | 13,6 | 4,3 |
| 10 | 5441 | 1,97 | 18,56 | 91,27 | 270,8 | 6,0 | 2,2 |
|   | 5445 | 2,03 | 20,09 | 95,44 | 284,0 | 6,2 | 2,2 |
|   | 5447 | 1,48 | 16,23 | 94,71 | 312,5 | 8,7 | 2,8 |

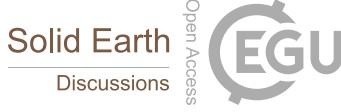

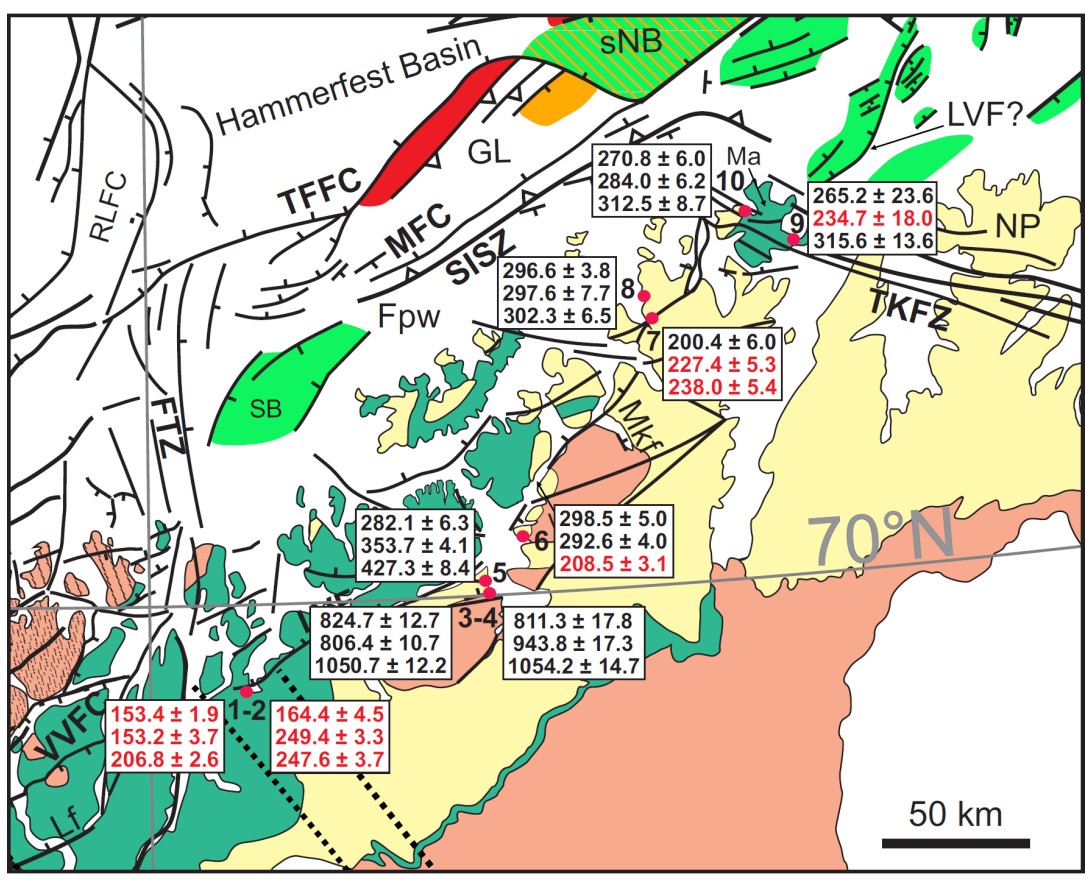

**Figure 4: Structural map of the study area showing the obtained K/Ar ages for each sample in Ma, including from top to bottom finest, intermediate and coarsest grainsize fraction ages. Ages in red are considered erroneous (cf. main text). Map modified after Indrevær et al., (2013) and Koehl et al. (2018). Legend and abbreviations as in Figure 1.**





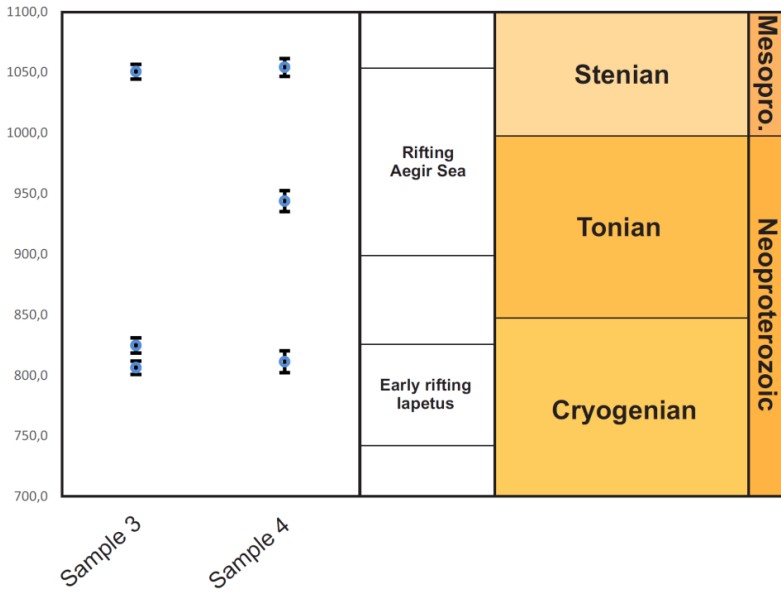

**Figure 5: Graph displaying the obtained Precambrian ages for brittle faults in the Alta-Kvænangen tectonic window. The vertical axis is time in Ma and the horizontal axis shows the dated sample numbers, which locations are displayed in Figure 1 & Figure 4. Columns to the right show extensional tectonic events related to the formation of the NW Baltoscandian basins during the opening of the Asgard Sea (Siedlecka et al., 2004; Nystuen et al., 2008; Cawood et al., 2010; Cawood & Pisarevsky, 2017) and to the earliest phase of rifting of the Iapetus Ocean-Ægir Sea (Li et al., 1999, 2008; Torsvik & Rehnström, 2001; Hartz & Torsvik, 2002), and associated geological Periods and Eras.**



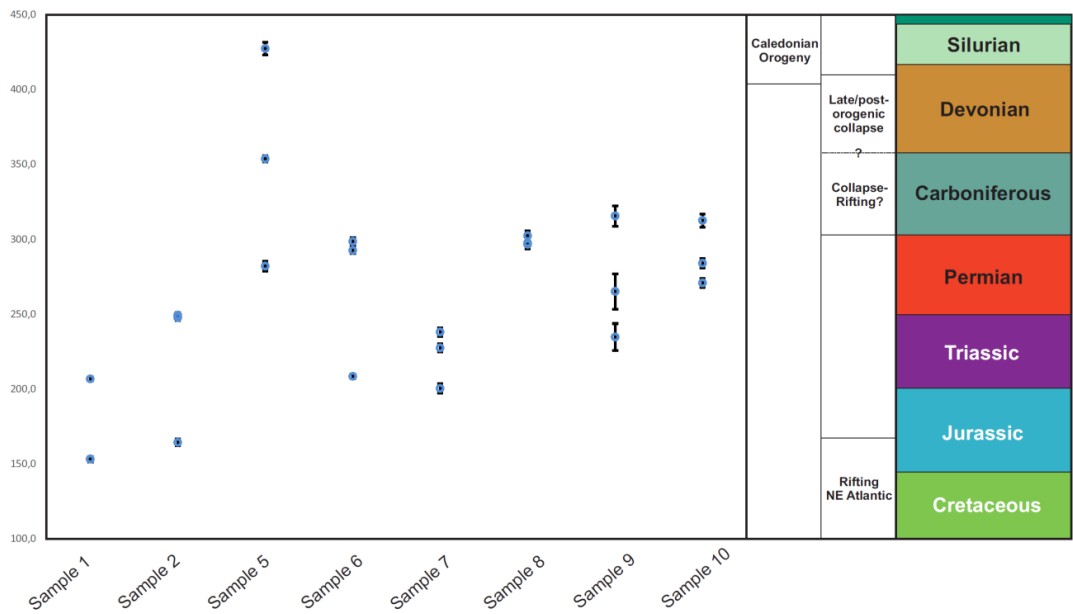

**Figure 6: Graph showing the Phanerozoic K/Ar ages and associated error obtained for fault gouge samples in NW Finnmark. The vertical axis represents time in Ma and the horizontal axis shows the dated sample numbers, which locations are displayed in Figure 1 & Figure 4. Columns to the right of the graph display major contractional (based on Ramberg et al., 2008; Vetti, 2008; Corfu et al., 2014) and extensional events (Faleide et al., 1993; Steltenpohl et al., 2011; Koehl et al., 2018) that affected North Norway in late Paleozoic-Mesozoic times, and associated geological time Periods.**









**Appendix A: X-Ray Diffraction spectrum of copper graphs showing the mineralogical composition of the dated fault-rock samples. Green, blue and red lines respectively represent coarse, intermediate and fine grainsize fractions for each sample. Abbreviations: chl = chlorite; ill = illite; Kfs = K-feldspar; lmt = laumontite; qz = quartz; sme = smectite.**