# Peer review of "Neoproterozoic and post-Caledonian exhumation and shallow faulting in NW Finnmark from K–Ar dating and p/T analysis of fault-rocks"

_Solid Earth, 2018_

## Referee Comment (RC1) · Anonymous Referee #1 · 14 Apr 2018

The paper by Koehl et al., presents the results of several K/Ar determinations of rocks in Finnmark Northernmost Norway. The results are placed in the context of a comprehensive structural evaluation. The work is generally well written and comprehensive. However, the text is excessively long and could reasonably be expected to be shorted to make the salient points more accessible to the readership. I do think this work has the potential to make a very useful contribution to the journal however I have some critical comments the authors may wish to consider and prepare a revision to address. The ages are distinct from other high temperature and Ar/Ar texturally constrained results from Finnmark which illustrate the importance of Caledonian processes. Much better evaluation of the results in the context of these other geochronometers is needed.
Specifically, why are the ages so different, is it simply due to different fabrics being dated, a different closure temperature of the system and mineral, or are secondary alteration processes at play? It is trivial to address this question but is needed to place the work in the current regional context. The reported ages may very well be correct but then more detail is required to put them in the context of post Caledonian process (in the main). More evidence is required to demonstrate that the clay minerals all formed at the same time and that they have not been subsequently modified with later fluid movement on the "dated" structures. For example, more SEM and EBSD textural work would be a distinct advantage in addressing how many different fabrics are associated with each of the sampled structures. What is the potential of fine fraction feldspar modifying the K/Ar results? Is there evidence of significant fluid alteration on these structures? I am somewhat concerned with the number of references to unpublished works that are cited as "submitted" (e.g. Koehl et al., Davids et al.,). I do not think it acceptable to heavily rely on such currently unpublished work so I would suggest that a summary of the salient points in those unpublished works is presented in this paper so all the evidence for statements in this work is available to readers. Other minor points Line 16: sentence structure I think you mean "during the opening of".... Line 34: replace "which" with "whose" Line 50; you probably should mention the timing of collisional phases as constrained this region prior to discussing post-Caledonian extension. See Kirkland et al., 2006, The structure and timing of lateral escape during the Scandian Orogeny: A combined strain and geochronological investigation in Finnmark, Arctic Norwegian Caledonides, which discusses the constraint on the timing of the collisional phase (431-428 Ma). Line 65: you need to present the evidence or at least discuss (if it is already published) the evidence for Timanian deformation in northern Finnmark. More discussion on the rational for a 30'c/km geotherm is needed. There are some regional thermobarometry studies that point to the peak P-T conditions which may be relevant assist in placing constraints on the retrograde thermal pathway. There are some sections of the text that need rewritten, for example evaluating the results against an unpublished (e.g. submitted) model (e.g. TKFZ development) by the same
authors and coming to the conclusion that the previous unpublished model is wrong seems odd to me. You already know it doesn't work with your data. Line 105-115; I would have thought it relevant to discuss the results on basement metamorphism as provided by pseudosection thermobarometry, as it is likely to be some of the most accurate P-t constraints in the region and at least provides some constraints for subsequent processes. ć Gasser et al., 2015; D. Gasser, P. Jeřábek, C. Faber, H. Stünitz, L. Menegon, F. Corfu, M. Erambert, M.J. Whitehouse Behaviour of geochronometers and timing of metamorphic reactions during deformation at lower crustal conditions: phase equilibrium modelling and U–Pb dating of zircon, monazite, rutile and titanite from the Kalak Nappe Complex, northern Norway. ć Kirkland et al., 2016: C.L. Kirkland, T.M. Erickson, T.E. Johnson, M. Danišík, N.J. Evans, J. Bourdet, B.J. McDonald, Discriminating prolonged, episodic or disturbed monazite age spectra: An example from the Kalak Nappe Complex, Arctic Norway, Chemical Geology, Volume 424, 2016, Pages 96-110,.

---

## Referee Comment (RC2) · D. Roberts (Referee) · 23 Apr 2018

D. Roberts (Referee)

david.roberts@ngu.no

General comments:

This is an interesting contribution to the Proterozoic and Palaeozoic, geological and faulting development of northwestern Finnmark and north Troms, northern Norway. Over the last half century, this part of the Norwegian Caledonides has witnessed great advances in our knowledge of the basic geology through detailed bedrock mapping aided by simultaneous stratigraphical and sedimentological studies. Other research investigations have focused on the geochemistry and dating of mafic dykes, low-grade and high-grade metamorphism, provenance studies, and the petrology and dating of

plutonic complexes in the famous Seiland Igneous Province and Honningsvåg Igneous Suite. The principal faults, and their trends and general evolution, are also well known, as seen from the geological maps on different scales and from descriptions of faults and fault-rocks in several publications, in some cases aided by satellite-sensor (e.g. Landsat) imagery. What has been largely lacking, with just a couple of exceptions, until now, is a study of the mineralogy of gouges and cataclasites in the cores of the many mapped faults; and, moreover, an attempt to date the mineral species that reside in these diverse fault-rock products. The manuscript in question here, by Koehl and colleagues, is therefore a very welcome addition to our knowledge of faulting in this part of Norway; one might say, another small "brick in the wall" of northern Norwegian geoscience. I would support its publication, subject to comparatively minor revisions as set out below.

The manuscript is generally well written, but my ex-editorial eyes spied a mixture of British and American English, in this case mostly Am-Eng, though with a disregard for some aspects of Am-Eng syntax. This is a job for the Editor to sort out. I do not know the rules of Solid Earth on this matter. Perhaps there are no defined rules to help authors? But as a Euopean publication, SE is closer to Britain than the USA; and in this case all three authors are European. Personally, I do know that the Norwegian Journal of Geology follows British-English. Quite possibly the SE is open to both language variants as long as there is complete consistency in any one manuscript? In this Koehl et alia ms, however, there is no consistency.

Specific comments

Here, I list a variety of comments by reference either to page numbers or in most cases to actual lines as numbered in the manuscript. Firstly, however, there are couple of general points relating to small errors or inconsistencies that appear on almost every page.

Page 1, and throughout the ms. The temporal adjectives Early, Mid and Late should

be capitalised (e.g., Mid Neoproterzoic). The authors are inconsistent on this point.

Several pages. Composite adjectives should be hyphenised, e.g, greenschist-facies rocks, but not when writing . . . The rocks were metamorphosed in greenschist facies.

Page 1, line 12 (also p.9, line 273 and p. 10, line 286). The correct term is 'Raipas Supergroup' (there are several groups within the Raipas Supergroup).

Page 2, line 53. Write Late Devonian. Then add reference to Guise & Roberts 2002 (NGU Bulletin).

Page 3, line 67. " …...fault complex, the age of which is yet uncertain (Zw. . . .". Then on line 69 " . . . onto the eastern Finnmark Platform where …" Next, on line 87 "cross-cutting".

Page 4, line 102. After 1983, add reference to Torgersen et al. 2015Âÿ NJG Vol 95. Then, after 1985, add reference to the classical paper by Gayer et al. 1987, Trans.Royal Soc.Edinburgh. Note the spelling of Ramsay, not Ramsey.

Page 4, line 104. Elvevold was not the first to write about the SIP. Many publications by Robins and students/colleagues. Here you can add Robins & Gardner 1975, Earth Planet.Sci.Letters 26. Page 4, Line 105. Again, Robins should be mentioned, in front of Corfu. Add Robins 1998, Geol. Mag. 135, 231-244.

Page 4, Lines 108-110. Here you need to achieve a balance between the two main interpretations for the Kalak. In line 108, write . . . " . . .is thought by some workers to represent . . .". Then, add a new sentence after (Kirkland et al. 2008). "However, others have provided robust evidence for a Baltican origin (e.g., Roberts & Siedlecka 2012, Zhang et al. 2016)."

Page 4, line 111. "to amphibolite-facies . . . . composed of psammites, . . . " (Note – psammites are metamophosed sandstones, so the prefix 'meta' is not required.). Then on line 112, Ramsay, not Ramsey. On line 113, add Robins & Gardner 1975 before Elvevold. On line 118, add Robins 1998 before Corfu.

Page 5, line 125. You should add Lippard & Roberts 1987 before Bergh.

Page 5, line 142. Should Nasuti et al. 2015 be 2015a ? Line 149, shows not show; then on line 150 thrusts not thrust.

Page 6, line 156. " . . .are juxtaposed against amphibolite-facies . . . " Then, line 174 " compositions and with the results of . . ." On line 180, "brittle faults that we encountered . . ."

Page 7, line 193. The Ksienzyk reference is not in the References. The very last word on this line should be 'whose', not which.

Page 7, lines 206-207. "have been shown to be a sensitive . . . . . . .low-grade . . ." Then, the end of this sentence . . . . . "in clastic sedimentary rocks." These are not unconsolidated sediments but lithified sedimentary rocks.

Page 8, line 230. "Synkinematic illite commonly grows . . ." Do not use adverbs of time in cases like this – e.g often, frequently, occasionally. There are many such examples in this manuscript. Line 242, " . . thus causing coarse fractions . . . " Line 245 " and, thus, yield rebust . . . ."

Page 9, line 247. contrary, not contrarily. Next line . . . . "is unlikely". Next line temperatures. Line 256, counterparts should be one word. Line 272 Raipas Supergroup. Also on next page, line 286.

Page 10, line 294. "accommodated a vertical displacement of >5-6 m." Line 305/6 "The northern fault accommodates . . ."

Page 11, line 310 "Along strike to the northeast, we . . ." Line 312 "that cross-cuts arkosic psammites . . " Line 313 replace "made" by the word composed

Page 11, the subheading 4.1.3. This is incorrect and misleading. These brittle faults are nowhere near the TKFZ. Rephrase as follows – "Brittle faults on Porsanger Peninsula and Magerøya (samples 8, 9 & 10)"

Page 12. The very first sentence (lines 338-340) needs some rewriting, as follows –
" are widespread in the Kalak Nappe Complex on the Porsanger Peninsula and in the
Magerøy Nappe on the island of Magerøya (Figure 1; cf. Koehl et al., in prep.)." This
Koehl et al manus has NOT been submitted to NJG, at least not received by the editor
(I have checked this). Line 343, "Magerøy Nappe rocks on . . ." Line 348, "Farther
north, on Magerøya, sample 9 corresponds . . ."

Page 13, line 385. " . . . /metavolcanites, gabbros and mica schists." Line 397, " is
composed of"

Line 398 " , commonly reworked . . ."

Page 14, line 400. " In places, epidote-chlorite-bearing "

Page 15, line 437. " is composed of . . . " Line 458, part of this sentence is unclear —"
younger age in tendency compared to ..". Authors please note: "compare to" means to
liken one thing to another, whereas "compare with" indicates that one wishes to show
the differences between one thing and another.

Page 16, line 477. Delete the words "along the TKFZ" ! Lines 478-481, In this sentence
(and in several other sentences in the ms) you give what you say are three "ages" for
the different fractions. These are just numbers or dates. You then have to interpret
them as real ages of mineral-forming events.

Page 17, line 499. Delete the words "segment of the TKFZ". Line 502, "suggesting that
the fault . . ." Line 507, "associated with the "

Page 18, line 522. " hence yielding ages younger than the actual age of faulting . . ."
Lines 535-36, "However, a minor . . . . . . . . spectra of all . . . . . . .suggests that the
. . . ."

Page 19, line 576, " . . . 3c). Subsequently, basement rocks were exhumed to a
shallow . .. "

Page 21 , lines 623/4. Do not use Ma here. Instead write . . . . "10-75 m per mill. yrs."
and later, "10-100 m per mill. yrs." This crops up at many other places in the ms.

Page 22, line 652. "at depths between 3 and 10 km." Line 662, "along fault segments
of the LVF and faults parallel to the TKFZ, showing . . . ."

Page 23, line 676, "suggesting that the dominant . . . " Line 683 "WNW-ESE-trending
faults associated with the" Line 691 "along the LVF and parallel to the TKFZ . . ."

Page 24, line 711, "earliest indications of . . ." Line 717 "Silurian age may reflect input
from an inherited" Line 733 " of the LVF and faults parallel to the TKFZ outlined . . ."
On next page, line 743, Koehl et al., in prep.

Page 26, line 777, " at shallow depths between 1 and 3.5 km . . ."

Page 27, lines 816/17, "although quite commonly associated with large amounts of
chlorite along fault segments of the LVF (e.g., the Talvik fault and the Sørkjosen faults
. . ."

Page 28, line 831 " . . analyses of gouge from a fault parallelling the TKFZ in . . ."
Line 839, "indicating that exposed rocks in northern . . ." Line 856, "erosion with only a
limited . . ."

Page 29, line 880, "A reliable Mesozoic K/Ar age was obtained only for the . . . "

Page 30, line 887 Spelling . . . scarcity, not scarsity Line 898 in prep. , not submitted
Line 906, "conjugate strike-slip faults have been reported from NW Finnmark . . . ."
Line 908 This statement is incorrect. The Timanian contraction in the Late Neoprotero-
zoic was directed towards the SW, not along the strike of the orogen.. This has been
described in numerous publications.

On pages 31, 32 & 33 I have noted dozens of small errors; these will surely be picked
up by the editor.

Page 38, lines 1151-53, The area covered by this forthcoming manuscript (NOT yet

submitted) seems to be comparable to that of the present contribution. How much repetition will there be, one wonders? But perhaps the authors will be describing completely different faults? – and taken mostly from the theses of E.Bergø and H.Lea ?

Brief comment on the K/Ar dating

The results are very interesting and only the second such study of dating mineral growth in gouges and cataclasites in the northwestern part of Finnmark. Many numbers/dates are given, for the three different grain-size fractions, so it is up to the authors to explain why they interpret every single number as a true age. The authors mention just "illite", sensu lato, but there are several polymorphs of illite, e.g., 1M, 1Md and 2M1, so do they know for sure which polymorph they have been analysing?

The figures

All the figures appear reasonable and acceptable, except for the incorrect placing of the TKFZ in Figure 1 and Figure 4. This error MUST be corrected. I have corresponded with the first author on an earlier occasion about this matter (in another paper of his in Solid Earth), so he knows the facts. The TKFZ is described and defined from the Varanger Peninsula, just outside the eastern limits of Figures 1 and 4, but continues WNWwards through the isthmus on the south side of the Nordkinn Peninsula (marked as NP in Figures 1 and 4). That is where the authors should write in the acronym TKFZ (in smaller capital letters).

I have not had time to control the figure captions.

Technical points

My paper copy is littered with small corrections and additions, typing errors, punctuation errors, etc. However, I leave all these to the chief editor, or even one of the desk editors.

David Roberts, Professor Emeritus, NGU, 7491 Trondheim, Norway 19 April 2018

––––––––––––––––––––––––––––––––

---

## Author Comment (AC1) · 13 Jun 2018

Dear Prof. Roberts,

thank you very much for your input on the manuscript. Here is our response to your comments. We hope that the changes we implemented improve the shortcomings of the manuscript highlighted by your comments and suggestions. Please do not hesitate to contact us shall this not be the case for some of your comments.

1. Comments from Dr. Roberts

Comment 1: Page 1, and throughout the ms. The temporal adjectives Early, Mid and

[Figure]

Late should be capitalised (e.g., Mid Neoproterzoic). The authors are inconsistent on this point. Comment 2: Several pages. Composite adjectives should be hyphenised, e.g, greenschist-facies rocks, but not when writing . . . The rocks were metamorphosed in greenschist facies. Comment 3: Page 1, line 12 (also p.9, line 273 and p. 10, line 286). The correct term is 'Raipas Supergroup' (there are several groups within the Raipas Supergroup). Comment 4: Page 2, line 53. Write Late Devonian. Then add reference to Guise & Roberts 2002 (NGU Bulletin). Comment 5: Page 3, line 67. " : : :..fault complex, the age of which is yet uncertain (Zw. . . .". Then on line 69 " . . . onto the eastern Finnmark Platform where : : :" Next, on line 87 "cross-cutting". Comment 6: Page 4, line 102. After 1983, add reference to Torgersen et al. 2015Âÿ NJG Vol 95. Then, after 1985, add reference to the classical paper by Gayer et al. 1987, Trans.Royal Soc.Edinburgh. Note the spelling of Ramsay, not Ramsey. Comment 7: Page 4, line 104. Elvevold was not the first to write about the SIP. Many publications by Robins and students/colleagues. Here you can add Robins & Gardner 1975, Earth Planet.Sci.Letters 26. Page 4, Line 105. Again, Robins should be mentioned, in front of Corfu. Add Robins 1998, Geol. Mag. 135, 231-244. Comment 8: Page 4, Lines 108-110. Here you need to achieve a balance between the two main interpretations for the Kalak. In line 108, write . . . " . . .is thought by some workers to represent . . .". Then, add a new sentence after (Kirkland et al. 2008). "However, others have provided robust evidence for a Baltican origin (e.g., Roberts & Siedlecka 2012, Zhang et al. 2016)." Comment 9: Page 4, line 111. "to amphibolite-facies . . . . composed of psammites, . . . " (Note – psammites are metamophosed sandstones, so the prefix 'meta' is not required.). Then on line 112, Ramsay, not Ramsey. On line 113, add Robins & Gardner 1975 before Elvevold. On line 118, add Robins 1998 before Corfu. Comment 10: Page 5, line 125. You should add Lippard & Roberts 1987 before Bergh. Comment 11: Page 5, line 142. Should Nasuti et al. 2015 be 2015a ? Line 149, shows not show; then on line 150 thrusts not thrust. Comment 12: Page 6, line 156. " . . .are juxtaposed against amphibolite-facies . . . " Then, line 174 " compositions and with the results of . . ." On line 180, "brittle faults that we encountered . . ."

[Figure]

Comment 13: Page 7, line 193. The Ksienzyk reference is not in the References. The very last word on this line should be 'whose', not which. Comment 14: Page 7, lines 206-207. "have been shown to be a sensitive . . . . . . . .low-grade . . ." Then, the end of this sentence . . . . . "in clastic sedimentary rocks." These are not unconsolidated sediments but lithified sedimentary rocks. Comment 15: Page 8, line 230. "Synkinematic illite commonly grows . . ." Do not use adverbs of time in cases like this – e.g often, frequently, occasionally. There are many such examples in this manuscript. Line 242, " . . thus causing coarse fractions . . . " Line 245 " and, thus, yield rebust . . . ." Comment 16: Page 9, line 247. contrary, not contrarily. Next line . . . . "is unlikely". Next line temperatures. Line 256, counterparts should be one word. Line 272 Raipas Supergroup. Also on next page, line 286. Comment 17: Page 10, line 294. "accommodated a vertical displacement of >5-6 m." Line 305/6 "The northern fault accommodates . . ." Comment 18: Page 11, line 310 "Along strike to the northeast, we . . ." Line 312 "that cross-cuts arkosic psammites . . " Line 313 replace "made" by the word composed Comment 19: Page 11, the subheading 4.1.3. This is incorrect and misleading. These brittle faults are nowhere near the TKFZ. Rephrase as follows – "Brittle faults on Porsanger Peninsula and Magerøya (samples 8, 9 & 10)" Comment 20: Page 12. The very first sentence (lines 338-340) needs some rewriting, as follows – " are widespread in the Kalak Nappe Complex on the Porsanger Peninsula and in the Magerøy Nappe on the island of Magerøya (Figure 1; cf. Koehl et al., in prep.)." This Koehl et al manus has NOT been submitted to NJG, at least not received by the editor (I have checked this). Line 343, "Magerøy Nappe rocks on . . ." Line 348, "Farther north, on Magerøya, sample 9 corresponds . . ." Comment 21: Page 13, line 385. " . . . /metavolcanites, gabbros and mica schists." Line 397, " is composed of" Comment 22: Line 398 " , commonly reworked . . ." Comment 23: Page 14, line 400. " In places, epidote-chlorite-bearing " Comment 24: Page 15, line 437. " is composed of . . . " Line 458, part of this sentence is unclear —" younger age in tendency compared to ..". Authors please note: "compare to" means to liken one thing to another, whereas "compare with" indicates that one wishes to show the differences

between one thing and another. Comment 25: Page 16, line 477. Delete the words "along the TKFZ" ! Lines 478-481, In this sentence (and in several other sentences in the ms) you give what you say are three "ages" for the different fractions. These are just numbers or dates. You then have to interpret them as real ages of mineral-forming events. Comment 26: Page 17, line 499. Delete the words "segment of the TKFZ". Line 502, "suggesting that the fault . . ." Line 507, "associated with the " Comment 27: Page 18, line 522. " hence yielding ages younger than the actual age of faulting . . ." Lines 535-36, "However, a minor . . . . . . . . . spectra of all . . . . . . .suggests that the. . . ." Comment 28: Page 19, line 576, " . . . 3c). Subsequently, basement rocks were exhumed to a shallow . .. " Comment 29: Page 21 , lines 623/4. Do not use Ma here. Instead write . . . . "10-75 m per mill. yrs." and later, "10-100 m per mill. yrs." This crops up at many other places in the ms. Comment 30: Page 22, line 652. "at depths between 3 and 10 km." Line 662, "along fault segments of the LVF and faults parallel to the TKFZ, showing . . . ." Comment 31: Page 23, line 676, "suggesting that the dominant . . . " Line 683 "WNW-ESE-trending faults associated with the" Line 691 "along the LVF and parallel to the TKFZ . . ." Comment 32: Page 24, line 711, "earliest indications of . . ." Line 717 "Silurian age may reflect input from an inherited" Line 733 " of the LVF and faults parallel to the TKFZ outlined . . ." On next page, line 743, Koehl et al., in prep. Comment 33: Page 26, line 777, " at shallow depths between 1 and 3.5 km . . ." Comment 34: Page 27, lines 816/17, "although quite commonly associated with large amounts of chlorite along fault segments of the LVF (e.g., the Talvik fault and the Sørkjosen faults . . ." Comment 35: Page 28, line 831 " . . analyses of gouge from a fault parallelling the TKFZ in . . ." Line 839, "indicating that exposed rocks in northern . . ." Line 856, "erosion with only a limited . . ." Comment 36: Page 29, line 880, "A reliable Mesozoic K/Ar age was obtained only for the . . . " Comment 37: Page 30, line 887 Spelling . . . scarcity, not scarsity Line 898 in prep. , not submitted Line 906, "conjugate strike-slip faults have been reported from NW Finnmark . . . ." Line 908 This statement is incorrect. The Timanian contraction in the Late Neoproterozoic was directed towards the SW, not along the strike of the orogen.. This has been described

in numerous publications. Comment 38: On pages 31, 32 & 33 I have noted dozens of small errors; these will surely be picked up by the editor. Comment 39: Page 38, lines 1151-53, The area covered by this forthcoming manuscript (NOT yet submitted) seems to be comparable to that of the present contribution. How much repetition will there be, one wonders? But perhaps the authors will be describing completely different faults? – and taken mostly from the theses of E.Bergø and H.Lea ? Comment 40: The results are very interesting and only the second such study of dating mineral growth in gouges and cataclasites in the northwestern part of Finnmark. Many numbers/dates are given, for the three different grain-size fractions, so it is up to the authors to explain why they interpret every single number as a true age. The authors mention just "illite", sensu lato, but there are several polymorphs of illite, e.g., 1M, 1Md and 2M1, so do they know for sure which polymorph they have been analysing? Comment 41: All the figures appear reasonable and acceptable, except for the incorrect placing of the TKFZ in Figure 1 and Figure 4. This error MUST be corrected. I have corresponded with the first author on an earlier occasion about this matter (in another paper of his in Solid Earth), so he knows the facts. The TKFZ is described and defined from the Varanger Peninsula, just outside the eastern limits of Figures 1 and 4, but continues WNWwards through the isthmus on the south side of the Nordkinn Peninsula (marked as NP in Figures 1 and 4). That is where the authors should write in the acronym TKFZ (in smaller capital letters).

2. Author's response

Comment 1: The temporal adjective should exclusively be capitalized when referring to an actual Stage or Period. However, this is not the case of early, mid and late Neoproterozoic, which official names on the Geological Time Scale 2012 are Tonian, Cryogenian and Ediacaran respectively. The same goes for early and late Carboniferous, which formal names are Mississippian and Pennsylvanian. On the contrary, Early, Middle and Late Devonian should be capitalized. Comment 2: Agreed. Comment 3: Agreed. Comment 4: Agreed. However, we believe that the addition of "Late"

[Figure]

ahead of "Devonian" is not needed since it is still debated when post-Caledonian extension initiated in the northern Norway, (e.g., Steltenpohl et al., 2011; Koehl et al., 2018). Comment 5: Agreed. Comment 6: Agreed. Unfortunately, we cannot include the suggested "Gayer et al. 1987" publication, because this scientific contribution is not accessible to the authors of the present manuscript. Moreover, the reference does not significantly improve the manuscript. Comment 7: Agreed. Comment 8: Agreed. Comment 9: Agreed. Comment 10: Agreed. Comment 11: Agreed for "thrust" and "show". However, only one Nasuti et al. (2015) study is used as supporting literature in the present manuscript and it is therefore not necessary to change reference to "Nasuti et al., 2015a". Comment 12: Agreed. Comment 13: Agreed. Comment 14: Agreed. Comment 15: Agreed. Comment 16: Agreed. Comment 17: Agreed. Comment 18: Agreed. Comment 19: Recent work (Koehl, 2018; Koehl et al., 2018; Koehl et al., submitted) interpret margin-oblique, WNW-ESE striking brittle faults on the island of Magerøya as fault segments of the Trollfjorden-Komagelva Fault Zone. Thus, we prefer to leave subheading 4.1.3 as it is now. Comment 20: Agreed that the sentence needs rewriting but not as suggested. The rewriting suggested by the referee does not reconcile the occurrence of units belonging to the Magerøy Nappe on the Porsanger Peninsula (Gasser et al., 2015). The authors agree with all the other remarks in comment 20 and updated the manuscript accordingly. Comment 21: Agreed. Comment 22: Agreed. Comment 23: Agreed. Comment 24: Agreed. Comment 25: Agreed. The data presented is called "ages" because they were calculated from isotopic ratios generated by radioactive decay. The discussion has to specify the interpretation of the age, whether it is crystallization, recrystallization, cooling or a geologically meaningless mixing age. Comment 26: Disagree with the referee's comment based on work by Koehl et al. (in review, Norwegian Journal of Geology). This work interprets WNW-ESE striking faults on Magerøya as fault segments of the TKFZ. The authors agree with the other remarks of the comment. Comment 27: Agreed. Comment 28: Agreed. Comment 29: Agreed. Comment 30: Agreed with the first remark but not with the second one (again see Koehl et al., 2018, submitted and Koehl, 2018) Comment 31: Agree

with first remark, but disagree with last two remarks (cf. Koehl et al., 2018, submitted and Koehl, 2018). Comment 32: Agreed. However, the authors do not agree with the last two remarks: the work by Koehl et al. is now submitted to the Norwegian Journal of Geology and is currently being reviewed. Comment 33: Agreed. Comment 34: Agreed. Comment 35: Disagree with first remark (cf. Koehl et al., submitted, also in Koehl, 2018), but agree with last two remarks. Comment 36: Agreed. Comment 37: Agreed. Comment 38: Agreed. Comment 39: The manuscript referred has now been submitted. Potential repetition in the other manuscript do not affect the review process of the present manuscript. Most faults from Lea 2016 and Bergø 2016 discussed with different perspectives than in Koehl et al. (submitted). Comment 40: Agreed. Analytical data presented here as ages cannot be treated like high-precision ages from e.g., modern U-Pb zircon dating. Our ages point to a time interval, which can be in some cases much larger than the analytical error given in the 2- error interval. We have to add this statement in the beginning of the discussion. Indeed, it would have been a great help for interpretation to identify and quantify the amounts of illite-polytyps 1Md, 1M and 2M1. However, since the mineralogical composition is dominated by smectite and chlorite the specific peaks for the polytyps are not recognizable due to peak overlap. Comment 41: Agreed.

3. Changes implemented

Comment 1: none. Comment 2: hyphenized "greenschist-facies" through manuscript where needed (eight occurrences). Comment 3: changed "Raipas Group" to "Raipas Supergroup" three times through manuscript. Comment 4: added "Guise & Roberts, 2002" in main text and to the reference list. Comment 5: changed as suggested by referee. Also changed "Finnmark Platform east" to "eastern Finnmark Platform" lines 140 and 149, and "Finnmark Platform west" to "western Finnmark Platform". Comment 6: added reference to Torgersen et al. (2015a) line 105 and to reference list. Changed other reference to "Torgersen et al. (2015)" to "Torgersen et al. (2015b)". Corrected "Ramsey" to "Ramsay" (two occurrences) Comment 7: added following references to

the main text and reference list: Robins and Gardner (1975); Robins (1998). Comment 8: added "by some workers" line 113. Also added "However, others have provided robust evidence for a Baltican origin (Roberts, 2007; Zhang et al., 2016)" lines 115-116, and both references (Roberts, 2007; Zhang et al., 2016) to reference list. Comment 9: changed "meta-psammite" and "meta-psammitic" to "psammite" and "psammitic" respectively (four occurrences in main text). Comment 10: added suggested reference line 132. Comment 11: changed "show" into "shows" line 156 and "thrust" into "thrusts" line 157. Comment 12: changed "with amphibolite facies" to "against amphibolite-facies" line 163 and "zeolite facies" and "prehnite-pumpellyite facies" to "zeolite-facies" and "prehnite–pumpellyite-facies" (multiple occurrences). Also changed "composition and with the result" to "compositions and with the results" line 181, and added "that" line 187. Comment 13: added Ksienzyk et al., 2016 to reference list and changed "which" to "whose" line 200. Comment 14: added "been" and changed "low grade" to "low-grade" line 214, and changed "sediments" to "sedimentary rocks" line 215. Comment 15: changed "often" to "commonly" lines 137, 237, 387, 408, 415, 433, 441, 575, 668, and 688; replaced "often" by "generally" line 420 and 839; changed "occasionally" to "in places" line 410 and 577; changed "are commonly undeformed and often appear" into "commonly are undeformed and appear" line 684; deleted "often" line 389; changed "leading" to "causing" line 249; replaced "provide with" by "yield" line 252. Comment 16: changed "contrarily" to "contrary" line 254; changed "are" into "is" line 255; changed "temperature" to "temperatures" line 256; changed "counter-parts" to "counterparts" line 266. Comment 17: changed sentence to "The northern fault accommodates" line 315-316. Comment 18: changed "northeastward" to "to the northeast" line 320; changed "crosscuts" to "cross-cuts" line 322; replaced "made" by "composed" line 323. Comment 19: none. Comment 20: changed sentence to "are widespread within units of the Magerøy Nappe on the Porsanger Peninsula and on the island" lines 348-349. However, changed "unit" to "rocks" as suggested by the referee line 353. Also added "on Magerøya, " line 358, and referece to "Robins, 1998" line 360. Comment 21: changed "metavolcanics" to "metavolcanites" and "mi-

caschists" to "mica schists" line 396, and "made" into "composed" line 408. Comment 22: changed "often" into "commonly" line 409. Comment 23: sentence updated to "In places, epidote–chlorite-bearing" line 411-412. Comment 24: changed "made" to "composed" line 450; deleted "in tendency" and changed "compared to" to "compared with" line 471. Comment 25: changed sentence line 477 to "Most dated fault gouges from segments of the LVF and TKFZ in the Kalak Nappe Complex and Magerøy Nappe on the Porsanger Peninsula and Magerøya yielded late Paleozoic ages"; Also added ", and we interpret them both as syn-kinematic crystallization along an active normal fault" lines 468-469; "syn-kinematic crystallization of authigenic illite during" lines 480-481; "are interpreted as syn-tectonic crystallization ages and" line 491; "syn-kinematic crystallization" lines 495-496; "These syn-kinematic crystallization ages suggest" line 506; ", which we all interpret as syn-tectonic crystallization ages. These ages suggest that" lines 513-514; "The ages obtained for the coarse and fine fractions are interpreted to represent syn-kinematic crystallization of authigenic illite." Lines 525-527; "syn-kinematic crystallization during" line 533; ", which we interpreted as syn-kinematic crystallization ages" lines 603-605 ; ", which we interpreted as crystallization ages" line 610; "crystallization age of" lines 732-733; "and interpreted as syn-kinematic crystallization ages" line 744; "syn-kinematic crystallization" lines 755-756; "syn-tectonic crystallization" line 774; "syn-kinematic crystallization" line 780; "syn-kinematic crystallization" line 863; "syn-tectonic crystallization" line 904. Comment 26: added reference to "Koehl et al., submitted" lines 514-515. Also changed "to" into "with" line 523. Comment 27: changed sentence line 539 to "hence yielding ages younger than the actual age", and line 553-554 to "However, a minor K-feldspar content observed in the diffraction spectra of all three grainsize fractions of both of these faults suggests that the" as suggested by the referee. Comment 28: changed "Then" to "Subsequently" line 596, and added "a" line 397. Comment 29: changed "Ma" into "Myr" (multiple occurrences through manuscript). Comment 30: changed sentence line 676 to "at depths between 3 and 10 km". Comment 31: added "that" line 700. Comment 32: replaced "evidence" by "indications" line 735; changed sentence line 743 to "Silurian age may reflect input

from an inherited". Comment 33: changed "–" to "and". Comment 34: changed sentence line 843 to "although generally associated with large amounts". Comment 35: changed "outcropping" to "exposed" line 866, and added "only a" line 884. Comment 36: changed sentence line 909 to "A reliable Mesozoic K–Ar syn-tectonic crystallization age was obtained only for the". Comment 37: changed "may exist in" to "have been reported from" line 934. Comment 38: corrected minor errors in indicated pages. Comment 39: comment irrelevant to the present manuscript. No change. Comment 40: added "Nevertheless, we emphasize that the analytical data presented as ages cannot be treated like high-precision ages from, e.g., modern U–Pb zircon dating, but rather point to a time interval that can, in some cases, be much larger than the analytical error given in the 2- error interval" lines 251-254. Also added "Since the mineralogical composition is dominated by smectite and chlorite, the specific peaks for the different illite polytypes are not recognizable due to peak overlap" lines 438-440. Comment 41: changed location of acronym "TKFZ" to space suggested by Prof. Roberts.

Best regards, Jean-Baptiste

———————————————————

---

## Author Comment (AC2) · 13 Jun 2018

Dear Sir, Madam, thank you very much for your input on the manuscript. Here is our response to your comments. We hope that the changes we implemented improve the shortcomings of the manuscript highlighted by your comments and suggestions. Please do not hesitate to contact us shall this not be the case for some of your comments.

1. Comments from Anonymous referee

Comment 1: the text is excessively long and could reasonably be expected to be

shorted to make the salient points more accessible to the readership. Comment 2: The ages are distinct from other high temperature and Ar/Ar texturally constrained results from Finnmark which illustrate the importance of Caledonian processes. Much better evaluation of the results in the context of these other geochronometers is needed. Specifically, why are the ages so different, is it simply due to different fabrics being dated, a different closure temperature of the system and mineral, or are secondary alteration processes at play? It is trivial to address this question but is needed to place the work in the current regional context. The reported ages may very well be correct but then more detail is required to put them in the context of post Caledonian process (in the main). Comment 3: More evidence is required to demonstrate that the clay minerals all formed at the same time and that they have not been subsequently modified with later fluid movement on the "dated" structures. For example, more SEM and EBSD textural work would be a distinct advantage in addressing how many different fabrics are associated with each of the sampled structures. Comment 4: What is the potential of fine fraction feldspar modifying the K/Ar results? Is there evidence of significant fluid alteration on these structures? Comment 5: I am somewhat concerned with the number of references to unpublished works that are cited as "submitted" (e.g. Koehl et al., Davids et al.,). I do not think it acceptable to heavily rely on such currently unpublished work so I would suggest that a summary of the salient points in those unpublished works is presented in this paper so all the evidence for statements in this work is available to readers. Comment 6: Line 16: sentence structure I think you mean "during the opening of": : :. Line 34: replace "which" with "whose". Comment 7: Line 50; you probably should mention the timing of collisional phases as constrained this region prior to discussing post-Caledonian extension. See Kirkland et al., 2006, The structure and timing of lateral escape during the Scandian Orogeny: A combined strain and geochronological investigation in Finnmark, Arctic Norwegian Caledonides, which discusses the constraint on the timing of the collisional phase (431-428 Ma). Comment 8: Line 65: you need to present the evidence or at least discuss (if it is already published) the evidence for Timanian deformation in northern Finnmark. Comment 9: More

discussion on the rational for a 30'c/km geotherm is needed. There are some regional thermobarometry studies that point to the peak P-T conditions which may be relevant assist in placing constraints on the retrograde thermal pathway. Comment 10: There are some sections of the text that need rewritten, for example evaluating the results against an unpublished (e.g. submitted) model (e.g. TKFZ development) by the same authors and coming to the conclusion that the previous unpublished model is wrong seems odd to me. You already know it doesn't work with your data. Comment 11: Line 105-115; I would have thought it relevant to discuss the results on basement metamorphism as provided by pseudosection thermobarometry, as it is likely to be some of the most accurate P-t constraints in the region and at least provides some constraints for subsequent processes. âËŸA ′c Gasser et al., 2015; D. Gasser, P. JeËĞrábek, C. Faber, H. Stünitz, L. Menegon, F. Corfu, M. Erambert, M.J. Whitehouse Behaviour of geochronometers and timing of metamorphic reactions during deformation at lower crustal conditions: phase equilibrium modelling and U–Pb dating of zircon, monazite, rutile and titanite from the Kalak Nappe Complex, northern Norway. âËŸA ′c Kirkland et al., 2016: C.L. Kirkland, T.M. Erickson, T.E. Johnson, M. Danišík, N.J. Evans, J. Bourdet, B.J. McDonald, Discriminating prolonged, episodic or disturbed monazite age spectra: An example from the Kalak Nappe Complex, Arctic Norway, Chemical Geology, Volume 424, 2016, Pages 96-110.

2. Author's response

Comment 1: Agreed. Comment 2: The correlation with other high-temperature chronometers is clear, as they are dealing with higher closure temperatures (U–Pb in zircon, or K–Ar and Ar–Ar in muscovite), the data we present simply have to be younger. K–Ar and Ar–Ar cooling ages on biotite are of special interest, because they are interpreted to reflect the cooling below 300°C (McDougall & Harrison, 1999). This temperature marks the transition from ductile to brittle deformation (Tullis & Yund, 1977; Scholz, 1988; Hirth & Tullis, 1989). Therefore, oldest fault gouge ages could be in the range of biotite cooling ages, but should never be older. For example, we obtained

one Silurian age (427.3 ± 8.4 Ma) representing either mixing with host-rock (inherited muscovite/illite) or Silurian brittle faulting. Based on the dominant transport direction and kinematics along the Talvik fault (dip-slip top-south), we propose that the Silurian age reflects continued thrusting rather than lateral escape (as proposed in Kirkland et al., 2006 for orogeny-oblique thrusts). This age is consistent with U–Pb ages on titanite and pseudosection thermobarometry of Gasser et al. (2015) constraining Caledonian retrograde shearing at temperature < 550°C to Silurian times (440-420 Ma). To carry out investigations on other low-temperature chronometers like U–Th/He dating or fission track is beyond the scope of this manuscript, which rather focuses on the structural framework. As fault gouges are the weakest point in the crust, they are often reactivated by ongoing deformation. This can lead to repeated growth of clay mineral, yielding significant ages in different grain fractions or to meaningless mixing ages. Secondary alteration processes are unlikely as clay minerals are very stable in the shallow crust, but can never be excluded. The context of post-Caledonian processes as been widely discussed by several authors, onshore and offshore (e.g., Davids et al., 2013; Gudlaugsson et al., 1998; Indrevær et al., 2013; Torgersen et al., 2014; Ksienzyk et al., 2016). Davids, C., Wemmer, K., Zwingmann, H., Kohlmann, F., Jacobs, J. and Bergh, S. G.: K-Ar illite and apatite fission track constraints on brittle faulting and the evolution of the northern Norwegian passive margin, Tectonophysics, 608, 196-211, 2013. Gasser, D., Jerábek, P., Faber, C., Stünitz, H., Menegon, L., Corfu, F., Erambert, M. and Whitehouse, M. J.: Behaviour of geochronometers and timing of metamorphic reactions during deformation at lower crustal conditions: phase equilibrium modelling and U–Pb dating of zircon, monazite, rutile and titanite from the Kalak Nappe Complex, northern Norway, Journal of Metamorphic Geology, 33, 513-534, 2015. Gudlaugsson, S. T., Faleide, J. I., Johansen, S. E. and Breivik, A. J.: Late Palaeozoic structural development of the South-western Barents Sea, Marine and Petroleum Geology, 15, 73-102, 1998. Hirth, G. and Tullis, J.: The Effects of Pressure and Porosity on the Micromechanics of the Brittle-Ductile Transition in Quartzite, Journal of geophysical Research, 94, 17825-17838, 1989. Indrevær, K., Bergh, S. G., Koehl, J-B., Hansen,

J-A., Schermer, E. R. and Ingebrigtsen, A.: Post-Caledonian brittle fault zones on the hyperextended SW Barents Sea margin: New insights into onshore and offshore margin architecture, Norwegian Journal of Geology, 93, 167-188, 2013. Ksienzyk, A.K., Wemmer, K., Jacobs, J., Fossen, H., Schomberg, A.C., Süssenberger, A., Lünsdorf, N.K. & Bastesen, E. 2016. Post-Caledonian brittle deformation in the Bergen area, West Norway: results from K–Ar illite fault gouge dating, Norwegian Journal of Geology, 96, 1-29. McDougall, I. & Harrison, T. M. 1999. Geochronology and Thermochronology by the 40Ar/39Ar Method, 2nd Oxford University Press, New York. Scholz, C. H.: The brittle-plastic transition and the depth of seismic faulting, Geologische Rundschau, 77/1, 319-328, 1988. Torgersen E., G. Viola, H. Zwingmann and C. Harris. 2014. Structural and temporal evolution of a reactivated brittle-ductile fault – Part II: Timing of fault initiation and reactivation by K-Ar dating of synkinematic illite/muscovite. Earth Planet. Sci. Lett., vol. 407, pp. 221-233. Tullis, J. and Yund, R. A.: Experimental deformation of dry Westerly Granite, Journal of geophysical Research, 82, 36, 5705-5718, 1977. Comment 3: It would be nice to have all kinds of accompanying investigations, but this is not the focus of our studies. The overwhelming majority of fault gouge papers in literature is accepted without SEM and EBSD work. Comment 4: The influence of older feldspar leading to younger ages by the low closure temperature has been explained in the text. A significant fluid alteration of feldspar in the fault gouge structure would lead to the formation of sericite, which in many cases can be recognized easily by a well-crystallized 001-white mica peak. We did not find any indication for this in our x-ray patterns. Comment 5: The work by Davids et al. submitted is only briefly integrated to the study for comparison purposes and the conclusions of the present manuscript do not rely at all on the conclusions of Davids et al. submitted. Regarding the Koehl et al. (submitted) work, it is already available on Researchgate and on the data repository of the University of Tromsø as part of the PhD thesis of the main author (Koehl, 2018; paper 1). Thus, we find it acceptable to refer to this earlier work. Koehl, J-B. P.: Mid/Late Devonian-Carboniferous extensional faulting in Finnmark and the SW Barents Sea, Unpublished PhD Thesis, University of Tromsø, Norway, 2018.

Comment 6: Agreed. Comment 7: Agreed. Comment 8: Agreed. Comment 9: Agreed. See answer to comment 11. Comment 10: We do not know whether the model presented in Koehl et al. (submitted; Norwegian Journal of Geology) is wrong or right. However, we hereby present an alternative to this model and try to be critical with our own work. We believe that mentioning the model presented in Koehl et al. (submitted) is important to the follow up of the that study, and does not impede the clarity of the present contribution. Comment 11: Agreed for the Gasser et al. (2015) study. However, we do not think the proposed work of Kirkland et al. (2016) is suitable to discuss the exhumation of Paleoproterozoic basement rocks in Finnmark because the samples dated in this study are from younger rocks of the Kalak Nappe Complex. This work is not appropriate either to discuss the exhumation history of Caledonian rocks during post-Caledonian extension because the ages obtained are pre-Caledonian and do not yield any information about peak Caledonian metamorphism.

3. Changes implemented

Comment 1: Removed "cf." trough manuscript (23 occurrences); changed "top-to-the-" expressions to "top- " consistently through the whole manuscript ; deleted "Caledonian" line 717; deleted "/age" line 722; changed "top-to-the-south, brittle (-ductile?), Caledonian" to "top-south Caledonian brittle" line 722; deleted fault orientation, e.g., "NNE-SSW striking", lines 332, 340, 353, 365-366, 1424, 1425-1426, 1427, 1429, 1430, 1435, 1437. Comment 2: see answer to comment 11. Added Tullis & Yund (1977), Gasser et al. (2015) and Ksienzyk et al. (2016) to reference list. Comment 3: none. Comment 4: none. Comment 5: none. Comment 6: changed "which" by "whose" line 34. Comment 7: added "Near the end of Caledonian contraction, lateral escape initiated in a NE-SW direction, and this episode of deformation was constrained to ca. 431–428 Ma by U–Pb and Ar–Ar dating (Kirkland et al., 2005, 2006; Corfu et al., 2006)" lines 51-53, and Kirkland et al. 2006 to the reference list. Comment 8: added "Siedlecka & Siedlecki, 1967; Roberts, 1972; Siedlecka, 1975" as supporting literature for Timanian deformation in northern Finnmark. Comment 9: see answer to comment 11. Also, we added the following references to support the choice of a 30°C/km geothermal gradient: Bugge et al., 2002; Chand et al., 2008; Vadakkepuliyambatta et al., 2015. Comment 10: none. Comment 11: added "U–Pb ages on titanite from northern Troms provide a minimum estimate of ca. 440–420 Ma for retrograde (< 550°C) Caledonian shearing (Gasser et al., 2015)" line 108-110, and Gasser et al. 2015 to reference list. Also added ", which is consistent with pseudosection thermobarometry and U–Pb ages on titanite constraining retrograde Caledonian shearing < 550°C (i.e., < 18 km depth) in the Kalak Nappe Complex in northern Troms to 440–420 Ma (Gasser et al., 2015)" in discussion chapter, line 655-657 and "This is consistent with thermobarometry and U–Pb ages constraining Caledonian retrograde shearing at temperature < 550°C to the Silurian at ca. 440 –420 Ma (Gasser et al., 2015)." lines 728-730.

Best regards, Jean-Baptiste